# MedFuncta: A Unified Framework for Learning Efficient Medical Neural Fields

**Paul Friedrich**[1] (ID)                PAUL.FRIEDRICH@UNIBAS.CH
**Florentin Bieder**[1] (ID)              FLORENTIN.BIEDER@UNIBAS.CH
**Julian McGinnis**[2] (ID)                JULIAN.MCGINNIS@TUM.DE
**Julia Wolleb**[3] (ID)                  JULIA.WOLLEB@YALE.EDU
**Daniel Rueckert**[2,4] (ID)             DANIEL.RUECKERT@TUM.DE
**Philippe C. Cattin**[1] (ID)            PHILIPPE.CATTIN@UNIBAS.CH

[1] *Department of Biomedical Engineering, University of Basel, Switzerland*

[2] *Institute of Artificial Intelligence in Medicine, Technical University of Munich, Germany*

[3] *Department of Biomedical Informatics and Data Science, Yale University, USA*

[4] *Department of Computing, Imperial College London, UK*

**Editors:** Accepted for publication at MIDL 2026

## Abstract

Research in medical imaging primarily focuses on discrete data representations that poorly scale with grid resolution and fail to capture the often continuous nature of the underlying signal. Neural Fields (NFs) offer a powerful alternative by modeling data as continuous functions. While single-instance NFs have successfully been applied in medical contexts, extending them to large-scale medical datasets remains an open challenge. We therefore introduce **MedFuncta**, a unified framework for large-scale NF training on diverse medical signals. Building on Functa, our approach encodes data into a unified representation, namely a 1D latent vector, that modulates a shared, meta-learned NF, enabling generalization across a dataset. We revisit common design choices, introducing a non-constant frequency parameter $\omega$ in widely used SIREN activations, and establish a connection between this $\omega$-schedule and layer-wise learning rates, relating our findings to recent work in theoretical learning dynamics. We additionally introduce a scalable meta-learning strategy for shared network learning that employs sparse supervision during training, thereby reducing memory consumption and computational overhead while maintaining competitive performance. Finally, we evaluate MedFuncta across a diverse range of medical datasets and show how to solve relevant downstream tasks on our neural data representation. To promote further research in this direction, we release our code, model weights and the first large-scale dataset - **MedNF** - containing $> 500\,\mathrm{k}$ latent vectors for multi-instance medical NFs. The project page is available at: https://pfriedri.github.io/medfuncta-io.

**Keywords:** Generalizable Neural Fields, Implicit Neural Representations, Meta-Learning

## 1. Introduction

It is a common choice to represent data on discretized grids, e.g., to represent an image as a grid of pixels. While this data representation is widely explored, it poorly scales with grid resolution and ignores the often continuous nature of the underlying signal (Dupont et al., 2022b). Recent research demonstrated that NFs provide an interesting, continuous alternative to represent different kinds of data modalities like sound (Sitzmann et al., 2020b), images (Stanley, 2007), shapes (Mescheder et al., 2019), videos (Chen et al., 2022), or 3D

scenes (Mildenhall et al., 2021), by treating data as neural functions that take spatial or temporal positions (e.g., pixel coordinates) as input and output the appropriate measurements (e.g., image intensity values). A detailed review of single-instance and generalizable NFs can be found in Section A. Single-instance NF training typically involves overfitting a neural network to a single signal. While this training paradigm yields accurate representations for single instances, it is prohibitively expensive when scaled to large datasets. This scalability issue has gained particular importance as researchers increasingly explore using NFs as compressed dataset representations (Dupont et al., 2022a; Schürholt et al., 2022a; Ma et al., 2024), where the neural network weights themselves are treated as a data modality. Naively training single-instance NFs additionally leads to highly unordered weight spaces across separately trained networks, which complicates downstream learning on the weights. Although specialized architectures designed to handle the permutation symmetries inherent to the multilayer perceptrons (MLPs) that typically comprise NFs exist (Navon et al., 2023; Schürholt et al., 2024), alternative frameworks that avoid these issues in the first place can greatly simplify downstream learning. To overcome these challenges, this work introduces a framework, called **MedFuncta**, that generalizes medical NFs from isolated, single-instance models to dataset-level neural representations. The central idea, borrowed from *Functa* (Dupont et al., 2022b) and shown in Figure 1, is to meta-learn a shared neural representation across the dataset, in which each signal is represented by a unique, signal-specific parameter vector, also referred to as latent, that conditions a shared network. This structure enables the model to capture and reuse redundancies across dif-

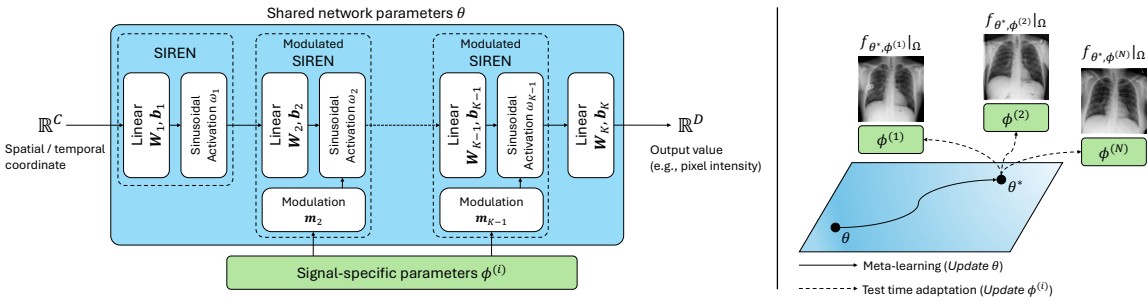

Figure 1: *(Left)* The proposed network with shared parameters $\theta$, that is conditioned by a single signal-specific parameter vector $\phi^{(i)}$. *(Right)* The proposed meta-learning strategy that, starting from a random initialization of $\theta$, learns shared network parameters $\theta^*$ in a way that we can fit a signal by updating $\phi^{(i)}$ for *few* steps.

ferent signals, drastically improving computational efficiency and scalability. Unlike prior methods that rely on patch-based representations (Dupont et al., 2022a; Bauer et al., 2023), our proposed framework represents each signal, from 1D time series to 3D volumetric data, with a single 1D latent vector. This abstraction enables *consistent downstream processing across diverse data types*, and is especially advantageous in medical applications, where the ability to *unify multiple data modalities under a common representation* is desirable (see Section B), and where the inherent capability of NFs to *handle irregularly sampled, hetero-*

*geneous data* provides further benefits. Our main contributions are threefold:

**(1) Optimization of Learning Dynamics at Scale:** We propose a non-constant, layer-dependent $\omega$-schedule for commonly used SIREN activations, significantly improving both convergence speed and reconstruction quality. We provide theoretical insights into the interplay between a layer's $\omega$-parameter and its *effective learning rate*, connecting these results to recent research on theoretical learning dynamics. **(2) Scalable Meta-Learning via Context Reduction:** We propose an efficient, context-reduced meta-learning framework to handle high-dimensional medical data. By utilizing sparse supervision during training, we significantly reduce memory consumption and computational overhead while maintaining competitive performance and speeding up the learning process. **(3) Comprehensive Evaluation and Open Resources** We demonstrate the versatility of MedFuncta across a diverse range of medical datasets and downstream tasks. To accelerate community research, we open-source our implementation, trained network weights, and a comprehensive dataset - **MedNF** - containing $> 500\,\mathrm{k}$ latent vectors for multi-instance NFs in medical imaging and machine learning.

## 2. Method

In this work, we aim to find parameter-efficient NFs for $N$ signals $\{s_1, \ldots, s_N : \mathbb{R}^C \to \mathbb{R}^D\}$, e.g., a set of time series, or images, by learning a functional representation of the signal $s_i$ given some context set $\mathcal{C}^{(i)} := \{(\mathbf{x}_j, \mathbf{y}_j)\}_{j=1}^M$ with $M$ coordinate-value pairs $(\mathbf{x}_j, \mathbf{y}_j) \in \mathbb{R}^C \times \mathbb{R}^D$. While parameterizing such a function as a neural network $f_\theta : \mathbb{R}^C \to \mathbb{R}^D$, with all parameters $\theta$ being optimized to fit a single signal is widely explored (Sitzmann et al., 2020b; Mildenhall et al., 2021; Saragadam et al., 2023), this approach is prohibitively expensive when scaled to large datasets.

**Network Architecture**  We argue that most sets of signals (datasets) contain large amounts of redundant information or structure that we can learn over the entire set. This is particularly true in medicine, where patients exhibit broadly similar yet slightly varying anatomies. We therefore define a neural network $f_{\theta, \phi^{(i)}} : \mathbb{R}^C \to \mathbb{R}^D$ with shared network parameters $\theta$ that represent this redundant information and additional signal-specific parameters $\phi^{(i)} \in \mathbb{R}^P$ that condition the base network to represent a specific signal $s_i$. We apply a $K$-layer MLP architecture with a hidden dimension of $L$ and FiLM modulated SIREN activations (Sitzmann et al., 2020b; Mehta et al., 2021), where all layers $k \in \{2, ..., K-1\}$ are defined as:

$$x \mapsto \sin\left(\omega_k\big(\underbrace{\mathbf{W}_k x + \mathbf{b}_k}_{\text{Linear}} + \underbrace{\mathbf{m}_k(\phi^{(i)})}_{\text{Modulation}}\big)\right), \tag{1}$$

with $\omega_k$ being the layer's frequency parameter, $\mathbf{W}_k$ and $\mathbf{b}_k$ being the weights and biases of the $k$-th layer, and $\mathbf{m}_k(\cdot)$ being a linear layer that maps the signal-specific parameters $\phi^{(i)}$ to a shift-modulation vector that is added in the base network's nonlinearity (Perez et al., 2018). The first layer is a SIREN layer that projects the input coordinate to a higher-dimensional space. The last layer is a linear layer that performs a simple mapping to the desired output dimension. An overview of the proposed architecture is shown in Figure 1.

**Network Initialization**  A proper initialization of NFs has been shown to have a huge influence on convergence and overall performance of the applied networks (Kania et al.,

2025; Yeom et al., 2025). We therefore initialize the network's weights and biases similar to Sitzmann et al. (2020b):

$$\mathbf{W}_k, \mathbf{b}_k \sim \mathcal{U}\left(-\frac{\sqrt{6/n}}{\omega_k}, \frac{\sqrt{6/n}}{\omega_k}\right), \tag{2}$$

with $n$ being the layer's input dimension. The first layer's weights and biases are initialized as $\mathbf{W}_1, \mathbf{b}_1 \sim \mathcal{U}(-1/n, 1/n)$.

**Introducing an $\omega$-Schedule**  While recent research treats $\omega$ as a single hyperparameter that remains constant over all network layers (Sitzmann et al., 2020b; Dupont et al., 2022b), we identify this as a main restriction when being applied in a generalization setting. We therefore propose to apply an **$\omega$-schedule** that linearly increases from $\omega_1$ to $\omega_K$ and find that this is equivalent to a *layer-wise learning rate schedule* that positively influences the network's learning dynamics. By carefully analyzing the interplay between a layer's $\omega$-parameter and it's learning rate $\tau$ (detailed derivation in Section E), we find that two layers with indices $m$ and $n$ and different $\omega$-values $\omega_m \neq \omega_n$ exhibit the following relation:

$$\frac{\tau_n}{\tau_m} = \left(\frac{\omega_m}{\omega_n}\right)^2. \tag{3}$$

This means that both layers show the same behavior, if we rescale the learning rates according to the inverse quadratic relationship $\tau \propto \frac{1}{\omega^2}$. This observation offers a so far overlooked perspective on the $\omega$-parameter in SIREN networks and establishes a connection to recent research on learning dynamics, providing a theoretical justification for introducing the proposed $\omega$-schedule. Chen et al. (2023) showed that shallow MLP layers yield faster convergence due to more informative gradients and a smoother loss landscape. They formalize this as the *layer convergence bias*, arguing that training strategies that prioritize low-frequency representations in shallow layers, while deferring high-frequency details to deeper layers, achieve better performance. They further find that shallow layers tolerate higher learning rates, whereas deeper layers begin to effectively learn once the learning rate decays. Our perspective on the $\omega$-parameter in SIRENs naturally fits into this framework. By gradually increasing $\omega$ with depth, we effectively lower the learning rate of deeper layers. This enforces a staged optimization dynamic, where shallow layers first stabilize around smooth, low-frequency features, and deeper layers subsequently refine high-frequency details. We validate this theoretical insight through ablation studies (Section 3), demonstrating that networks incorporating our proposed $\omega$-schedule outperform current state-of-the-art networks with a constant $\omega$-parameter.

**Efficient Meta-Learning with Context Reduction**  To efficiently create a set of NFs, we aim to meta-learn the shared parameters $\theta$ such that we can fit a signal $s_i$ by only optimizing $\phi^{(i)}$ for *very few* update steps (see Figure 1). We follow a CAVIA approach (Zintgraf et al., 2019), shown in Figure 2, by defining an optimization process over the shared model parameters:

$$\theta^* = \arg\min_\theta \frac{1}{N} \sum_{i=1}^{N} \mathcal{L}_{\mathrm{MSE}}(\phi_G^{(i)}, \theta; \mathcal{C}^{(i)}), \tag{4}$$

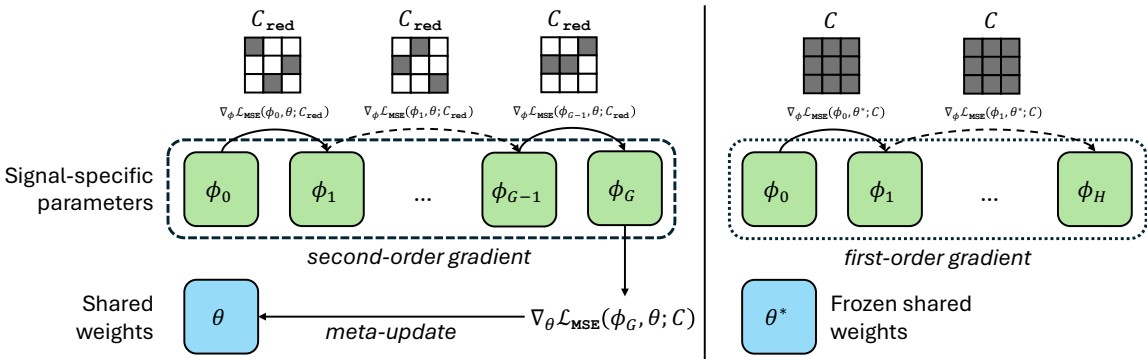

Figure 2: *(Left)* The proposed approach for meta-learning the shared model parameters $\theta$. An Algorithm describing the full meta-learning approach can be found in Section F. *(Right)* The proposed test time adaptation scheme.

where in each *meta/outer-loop* update step, the *inner-loop* optimizes $\phi^{(i)}$ from scratch $(\phi_0^{(i)} := \mathbf{0})$, performing $G$ update steps $\phi_{g+1}^{(i)} := \phi_g^{(i)} - \alpha \nabla_\phi \mathcal{L}_{\texttt{MSE}}(\phi_g^{(i)}, \theta, \mathcal{C}^{(i)})$, using stochastic gradient descent (SGD) with a fixed learning rate $\alpha$. The meta-update is performed using AdamW (Loshchilov and Hutter, 2019) with a learning rate $\beta$ that follows a cosine annealing learning rate schedule (Loshchilov and Hutter, 2016). All optimization steps aim to minimize the reconstruction error when evaluating the learned function $f_{\theta, \phi_g^{(i)}}$ on a given context set $\mathcal{C}^{(i)}$, by minimizing the mean squared-error (MSE) loss:

$$\mathcal{L}_{\texttt{MSE}}(\phi_g^{(i)}, \theta; \mathcal{C}^{(i)}) := \frac{1}{|\mathcal{C}^{(i)}|} \sum_{j \in \mathcal{C}^{(i)}} \|f_{\theta, \phi_g^{(i)}}(\mathbf{x}_j) - \mathbf{y}_j\|_2^2. \tag{5}$$

Performing a single meta-update step involves backpropagating through the entire inner-loop optimization, which requires retaining the computational graph in GPU memory to compute second-order gradients (Finn et al., 2017).[1] This resource-intensive task does not scale well to high-dimensional signals. While first-order approximations (Finn et al., 2017; Nichol et al., 2018) or auto-decoder training approaches that do not rely on second-order optimization exist (Park et al., 2019), recent research has shown that this results in severe performance drops or unstable training (Dupont et al., 2022b,a). To overcome this limitation, we propose to make use of a **reduced context set** $\mathcal{C}_{\texttt{red}}^{(i)}$ during the inner-loop optimization (Tack et al., 2023). This reduced context set contains a subset of the full context set $\mathcal{C}_{\texttt{red}}^{(i)} \leq \mathcal{C}^{(i)}$, thus saving GPU memory that is required for second-order optimization. We obtain the reduced context set by randomly sampling $\gamma |\mathcal{C}^{(i)}|$ coordinate-value pairs from $\mathcal{C}^{(i)}$. We empirically find that reducing the selection ratio $\gamma$ results in

---

1. Updating $\theta$ necessitates backpropagation through all inner-loop parameters $\phi_{1:G}$, each of which is itself a function of $\theta$. Consequently, computing the update for $\theta$ involves Hessian-vector products, which in turn demand storing the complete inner-loop computational graph. More information can be found in Section F.

marginal performance drops, while significantly reducing the required GPU memory and speeding up the training (see Table 4).

**Fitting Neural Fields at Test Time** Given the meta-learned model parameters $\theta^*$, we fit a NF to each signal $s_1, ..., s_N$, by optimizing the signal-specific parameter vectors $\phi^{(1)}, ..., \phi^{(N)}$. We start with initializing a signal-specific parameter vector $\phi^{(i)} := \mathbf{0}$ and optimize $\phi^{(i)}$ for $H$ steps by minimizing $\mathcal{L}_{\texttt{MSE}}(\phi^{(i)}, \theta^*; \mathcal{C}^{(i)})$. We do this for all $N$ signals. As no second-order optimization is required at test time (see Figure 2), we can make use of the full context set $\mathcal{C}^{(i)}$, i.e., we use all the available information at test time. A set of NFs representing the signals $s_1, ..., s_N$ is therefore defined by the network architecture, the shared model parameters $\theta^*$, and the signal-specific parameters $\phi^{(1)}, ..., \phi^{(N)}$. While meta-learning $\theta^*$ requires solving a complex optimization problem, fitting a NF at test time (i.e., optimizing $\phi^{(i)}$) simply requires $H$ SGD updates, which results in fast and low-resource inference ($< 0.5\,$s and $< 1\,$GB GPU memory for a $64 \times 64$ image), a desirable property in medical applications.

## 3. Experiments

**Datasets** We conduct experiments on a diverse set of publicly available datasets, spanning medical signals of different modalities, to demonstrate the flexibility of our proposed method: a single-lead `ECG` dataset (Kachuee et al., 2018), a `Chest X-ray` dataset (Wang et al., 2017), a `Pneumonia Chest X-ray` dataset (Kermany et al., 2018), a `Retinal OCT` dataset (Kermany et al., 2018), a `Fundus Camera` dataset (Liu et al., 2022), a `Dermatoscope` image dataset (Codella et al., 2019; Tschandl et al., 2018), a `Colon Histopathology` dataset (Kather et al., 2019), a `Cell Microscopy` dataset (Ljosa et al., 2012), a `Brain MRI` dataset (Baid et al., 2021; Bakas et al., 2017; Menze et al., 2014), and a `Lung CT` dataset (Armato III et al., 2011). No preprocessing was needed for `ECG`. For all 2D datasets, we use preprocessed versions from MedMNIST (Yang et al., 2021, 2023). The `Brain MRI` and `Lung CT` datasets were preprocessed as described by Friedrich et al. (2024).

**Implementation Details** All networks were trained with $G = 10$ inner-loop and $H = 20$ test time adaptation steps. The inner-loop learning rate was set to $\alpha = 10^{-2}$ and the outer-loop learning rate to $\beta = 3 \times 10^{-6}$. Network configurations and further training details are reported in Section C. All experiments were carried out on a single NVIDIA A100 (40 GB) GPU. Our implementation is publicly available at https://github.com/pfriedri/medfuncta.

**Reconstruction Quality** We first validate that our proposed approach can fit a wide range of medical signals by performing reconstruction experiments. We meta-learn the shared network parameters $\theta$ on a training set and evaluate the reconstruction quality on a hold-out test set. All models are trained for a fixed number of $250\,$k iterations, and testing is performed using the weights that achieve the best validation scores. We measure mean squared error (MSE), peak signal-to-noise ratio (PSNR), structural similarity index measure (SSIM), and learned perceptual image patch similarity (LPIPS) (Zhang et al., 2018) and report the results in Table 1. Qualitative examples of the performed reconstruction experiments are shown in Figure 3. While performance is generally better on homogeneous datasets, where redundancies can more effectively be exploited, the proposed method also

Table 1: Mean reconstruction quality of our proposed method, evaluated on a hold-out test set after meta-learning for 250 k iterations. MSE scores are multiplied by $10^3$. The spatial dimensions are 1D: 187, 2D: $64 \times 64$, 3D: $32 \times 32 \times 32$.

| Dim. | Dataset | MSE ($\downarrow$) | PSNR ($\uparrow$) | SSIM ($\uparrow$) | LPIPS ($\downarrow$) |
|---|---|---|---|---|---|
| 1D | ECG | 0.086 | 43.301 | 0.964 | – |
| 2D | Chest X-ray | 0.097 | 40.719 | 0.985 | 0.013 |
| | Pneumonia Chest X-ray | 0.146 | 39.301 | 0.977 | 0.014 |
| | Retinal OCT | 0.203 | 37.321 | 0.934 | 0.071 |
| | Fundus Camera | 0.054 | 43.151 | 0.978 | 0.006 |
| | Dermatoscope | 0.133 | 40.273 | 0.962 | 0.023 |
| | Colon Histopathology | 0.943 | 31.886 | 0.925 | 0.021 |
| | Cell Microscopy | 0.013 | 49.944 | 0.994 | 0.008 |
| 3D | Brain MRI | 0.130 | 39.191 | 0.993 | – |
| | Lung CT | 1.561 | 28.325 | 0.913 | – |

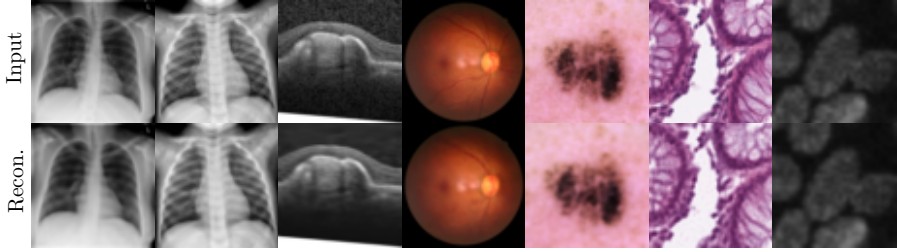

Figure 3: Input and reconstruction examples from the hold-out test set for *(from left to right)* `Chest X-ray`, `Pneumonia Chest X-ray`, `Retinal OCT`, `Fundus Camera`, `Dermatoscope`, `Colon Histopathology`, and `Cell Microscopy` images.

learns to represent complex inhomogeneous datasets, such as the `Colon Histopathology` dataset. Additional qualitative results can be found in Section J.

**Scaling MedFuncta to High-Resolution Signals**    To highlight our proposed approach's computational efficiency and scalability, we additionally evaluate its performance on higher-resolution signals. We, therefore, perform additional reconstruction experiments over multiple datasets, using images with a resolution of $128 \times 128$ and $224 \times 224$ as supervision signals. Reconstruction scores after 250 k meta-learning steps are reported in Table 2. Qualitative results are shown in Section J. The results demonstrate that our proposed method can reconstruct high-resolution signals, even when being trained on a single 40 GB GPU only. We believe that larger networks and longer, distributed training would further improve performance, especially on the $224 \times 224$ data, which runs under substantially different conditions due to hardware constraints.

**Classification Experiments**    To assess whether the learned representation captures relevant information about the underlying signal, we perform classification experiments on the signal-specific parameters $\phi$ (Dupont et al., 2022b; Navon et al., 2023), using a $k$-Nearest-Neighbor ($k$-NN) classifier, or a 3-layer MLP with ReLU activations and dropout. We

Table 2: Mean reconstruction quality of MedFuncta on higher resolutions. We use the setup from Section C, only changing batch size $B$, representation size $P$, selection ratio $\gamma$, $\omega_1 = 30$, and $\omega_K = 300$ . We also report the required training GPU memory in GB. MSE scores are multiplied by $10^3$.

| Dataset | Resolution | $B$ | $P$ | $\gamma$ | MSE ($\downarrow$) | PSNR ($\uparrow$) | SSIM ($\uparrow$) | Mem. ($\downarrow$) |
|---|---|---|---|---|---|---|---|---|
| Chest X-ray | $128 \times 128$ | 8 | 8192 | 0.25 | 0.216 | 37.174 | 0.952 | 28.68 |
| | $224 \times 224$ | 4 | 16384 | 0.10 | 0.401 | 34.510 | 0.909 | 25.39 |
| Dermatoscope | $128 \times 128$ | 8 | 8192 | 0.25 | 0.277 | 37.072 | 0.906 | 28.68 |
| | $224 \times 224$ | 4 | 16384 | 0.10 | 0.472 | 34.752 | 0.920 | 25.39 |

compare these simple classifiers on our MedFuncta representation to ResNet50 (He et al., 2016) and EfficientNet-B0 (Tan and Le, 2019) on the original data, and report the number of network parameters, training time, accuracy, and F1 scores. All models were trained for 50 epochs using AdamW with a learning rate of $10^{-3}$. The scores in Table 3 show the classification performance on a hold-out test set based on the model parameters yielding the highest validation accuracy. We find that solving the two classification tasks (binary

Table 3: Classification Performance. We report the number of network parameters, the training time in seconds, accuracy, as well as F1 scores.

| Dataset (*Classes*) | Classifier | Param. | Time ($\downarrow$) | Acc. ($\uparrow$) | F1 ($\uparrow$) |
|---|---|---|---|---|---|
| Pneumonia Chest X-ray (2) | $k$-NN ($k = 1$) on $\phi$ | 0 | 0 | 81.57 | 0.87 |
| | $k$-NN ($k = 3$) on $\phi$ | 0 | 0 | 80.93 | 0.87 |
| | MLP on $\phi$ | $1.2 \times 10^6$ | 45 | **89.10** | **0.88** |
| | ResNet50 | $23.5 \times 10^6$ | 450 | 83.49 | 0.80 |
| | EfficientNet-B0 | $4.0 \times 10^6$ | 270 | 84.46 | 0.82 |
| Dermatoscope (7) | $k$-NN ($k = 1$) on $\phi$ | 0 | 0 | 68.98 | 0.38 |
| | $k$-NN ($k = 3$) on $\phi$ | 0 | 0 | 69.28 | 0.32 |
| | MLP on $\phi$ | $1.2 \times 10^6$ | 65 | **74.96** | 0.48 |
| | ResNet50 | $23.5 \times 10^6$ | 700 | 74.36 | **0.49** |
| | EfficientNet-B0 | $4.0 \times 10^6$ | 410 | 70.42 | 0.44 |

and multi-class) on our proposed representation $\phi$ generally works well. We outperform both ResNet50 and EfficientNet-B0, applied to the original images, in terms of accuracy and can demonstrate competitive F1 scores, while requiring less training time and model parameters. These results indicate that our proposed representation actually captures informative features of the underlying signals. The observed performance improvement may be attributable to removing redundant signal components, which are in $\theta$ and not in $\phi$.

**Ablation Studies** To validate our proposed **context reduction strategy**, we study the effect of the context selection ratio $\gamma$ on the reconstruction quality. The results, presented in Table 4, demonstrate that reducing the context set in the inner-loop significantly reduces the required GPU memory while resulting in marginal performance drops. We identify a selection ratio of $\gamma = 0.25$ as a good trade-off. Compared to using the full context set, we reduce GPU memory usage to $\sim 30\%$ and cut the required training time by more than $50\%$, while incurring a marginal loss of less than $1\,\text{dB}$ in PSNR and 0.004 in SSIM. To assess the effectiveness of our proposed $\boldsymbol{\omega}$-**schedule**, we sweep over a combination of different

Table 4: The effect of the context selection ratio $\gamma$ on the reconstruction quality and GPU memory required for training. Measured on `Chest X-ray` dataset ($64 \times 64$) after $100\,\mathrm{k}$ iterations and a batch size of 12, using the baseline configuration. We also report the training time for 100 iterations, averaged over $50\,\mathrm{k}$ iterations following $2\,\mathrm{k}$ warm-up iterations.

| Selection Ratio ($\gamma$) | 0.1 | 0.25 | 0.5 | 0.75 | 1.0 |
|---|---|---|---|---|---|
| PSNR (dB) | 34.33 | 35.62 | 36.22 | 36.53 | 36.60 |
| SSIM | 0.941 | 0.955 | 0.958 | 0.959 | 0.959 |
| Memory [GB] | 6.77 | 11.43 | 19.32 | 28.66 | 34.34 |
| Time [s] / 100 iterations | 37.38 | 42.04 | 68.75 | 83.18 | 91.61 |

configurations with $\omega_1 = \{10, 20, 30, 40, 50\}$ and $\omega_K := \delta\omega_1$, with $\delta = \{1, 2, 5, 10, 20\}$. The results, shown in Figure 4, demonstrate that applying our proposed schedule consistently improves the performance over setups with a single $\omega$-parameter across all layers. We further observe a performance drop for large $\omega_K$-values, which can be attributed to training collapse. Reducing $\omega_K$ has proven effective in mitigating such instability, especially for high-dimensional signals. As a last experiment, we compare our baseline approach to Functa

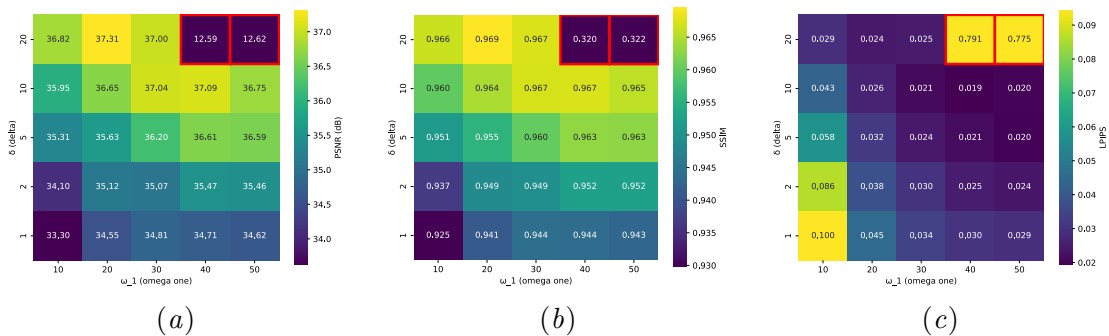

Figure 4: Grid search over different $\omega_1$ and $\delta$ parameters. We report ($a$) PSNR, ($b$) SSIM and ($c$) LPIPS after $25\,\mathrm{k}$ iterations. Outliers with red borders were excluded from color scaling. Measured on the `Chest X-ray dataset` ($64 \times 64$).

(Dupont et al., 2022b), two versions of COIN++ (Dupont et al., 2022a) (one working on full images and one operating on $32 \times 32$ image patches) and SpatialFuncta (Bauer et al., 2023). We additionally study the effect of introducing a global learning rate schedule, our proposed $\omega$-schedule, or the proposed context reduction scheme $\mathcal{C}_{\mathtt{red}}$. Implementation details for all comparing methods can be found in Section D. The results in Table 5 show that our approach outperforms Functa by approximately $\sim 6.4\,\mathrm{dB}$ PSNR and that each proposed component provides a consistent improvement over the baseline (we also consider $\mathcal{C}_{\mathtt{red}}$ as a practical improvement). We also surpass SpatialFuncta and both COIN++ variants, including the patched version that operates on $4\times$ fewer pixels. We believe that incorporating the findings from this paper into patch-based methods could further enhance their quality

Table 5: The effect of introducing a global learning rate schedule, our proposed $\omega$-schedule, or a reduced context set $\mathcal{C}_{\mathtt{red}}$ with $\gamma = 0.25$, as well as a combination of all of them in comparison to other approaches. Measured on `Chest X-ray` dataset $(64 \times 64)$ after $250\,\mathrm{k}$ training iterations. MSE scores are multiplied by $10^3$.

| Method | BS | Param. | Memory (GB) | MSE ($\downarrow$) | PSNR ($\uparrow$) | SSIM ($\uparrow$) | LPIPS ($\downarrow$) |
|---|---|---|---|---|---|---|---|
| Functa | 12 | $17.1 \times 10^6$ | 25.13 | 0.403 | 34.304 | 0.940 | 0.059 |
| COIN++ | 12 | $11.5 \times 10^6$ | 15.74 | 0.379 | 34.566 | 0.940 | 0.056 |
| SpatialFuncta | 12 | $2.1 \times 10^6$ | 10.69 | 0.587 | 32.782 | 0.915 | 0.149 |
| COIN++ (patched) | 12 | $4.5 \times 10^6$ | 4.29 | 0.153 | 38.477 | 0.976 | 0.014 |
| Ours (*Baseline*) | 12 | $7.7 \times 10^6$ | 34.34 | 0.179 | 37.836 | 0.970 | 0.016 |
| Ours $+$ *Global lr-sched* | 12 | $7.7 \times 10^6$ | 34.34 | 0.168 | 38.282 | 0.973 | 0.017 |
| Ours $+$ *$\omega$-Schedule* | 12 | $7.7 \times 10^6$ | 34.34 | 0.119 | 39.684 | 0.979 | 0.013 |
| Ours $+$ $\mathcal{C}_{\mathtt{red}}(\gamma = 0.25)$ | 24 | $7.7 \times 10^6$ | 21.51 | 0.205 | 37.338 | 0.968 | 0.019 |
| Ours $+$ *All* | 24 | $7.7 \times 10^6$ | 21.51 | **0.097** | **40.719** | **0.985** | **0.013** |

and scalability. Exploring patch-based representations, however, was beyond the scope of this work. Qualitative comparisons between all methods are provided in Section I.

## 4. Dataset: MedNF

While neural fields, or neural network weights in general, emerged as a novel data modality in the deep learning community, large-scale datasets like Implicit-Zoo (Ma et al., 2024) or Model-Zoo (Schürholt et al., 2022b) are scarce and not available for the medical domain. To promote research on weight space learning, finding well-performing architectures that operate on our representation, or solving further downstream tasks, we release a large-scale dataset, called **MedNF**, that contains seven sub-datasets - ChestNF, PathNF, DermaNF, OctNF, PneumoniaNF, RetinaNF, TissueNF - with a total of more than $500\,\mathrm{k}$ NFs. A detailed description of the datasets can be found in Section G. We release the dataset at https://doi.org/10.5281/zenodo.14898708.

## 5. Discussion

Motivated by recent progress in learning generalizable NFs, we present **MedFuncta**, a unified framework for efficiently learning large-scale neural representations of medical signals. We not only introduce a novel $\omega$-schedule for commonly used SIRENs, effectively enforcing a staged optimization process through layer-wise learning rates, but also address the computational overhead of previous methods by presenting an efficient meta-learning framework that utilizes training with sparse supervision. We validate our findings through extensive experiments and ablations across a wide range of diverse medical signals and demonstrate that we can solve relevant downstream tasks on our proposed neural representation. While this work is still limited to rather low resolutions, we believe that working on such neural representations holds substantial long-term potential, especially for heterogeneous medical data. For future research, we identify two main directions: (1) further scaling the method by exploring different architectures, conditioning mechanisms, or more advanced training approaches such as sequential mini-batch processing in the inner loop or novel optimiza-

tion strategies (McGinnis et al., 2025); (2) exploring further downstream tasks that benefit from generalizing NFs across a dataset. Possible applications include but are not limited to resolution-agnostic synthesis (Dupont et al., 2021b), image segmentation (Vyas et al., 2025), registration (Wolterink et al., 2022), atlas building (Dannecker et al., 2024), spatiotemporal growth modeling (Bieder et al., 2024), and various inverse problems like super-resolution (McGinnis et al., 2023). We hope that **MedFuncta** inspires the community and serves as a foundation for future research into neural representations of medical data.

## Acknowledgments

This work was financially supported by the Werner Siemens Foundation through the MIRACLE II project. JM is supported by Bavarian State Ministry for Science and Art (Collaborative Bilateral Research Program Bavaria – Québec: AI in medicine, grant F.4-V0134.K5.1/86/34). JW is funded by the Swiss National Science Foundation (Grant No. P500PT_222349)

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

## Appendix A. Related Work

**Neural Fields** Neural Fields (Xie et al., 2022) model signals as continuous neural functions that map from some domain to the corresponding signal value. Recent advances in NFs have mainly focused on alleviating the spectral bias (Rahaman et al., 2019) of commonly used MLPs. A variety of different strategies evolved, ranging from the application of Fourier Features in ReLU MLPs (Tancik et al., 2020), over periodic sinusoidal activation functions like SIREN (Sitzmann et al., 2020b), to using Gabor wavelet-based nonlinearities like WIRE (Saragadam et al., 2023), or multi-resolution hash-grid encodings like in InstantNGP (Müller et al., 2022). Research not only focused on network architectures and activation functions, but also explored training strategies like soft mining (Kheradmand et al., 2024), or context-pruning (Tack et al., 2023) that both aim to identify informative samples, as well as network initialization strategies (Yeom et al., 2025; Kania et al., 2025). In the medical field, single-instance NFs have become particularly relevant in settings where data is scarce, irregularly sampled (Huang et al., 2023), or should be free of cohort priors (McGinnis et al., 2023). Applications include slice-to-volume and unsupervised dynamic MR reconstruction (Xu et al., 2023; Feng et al., 2025), self-supervised motion correction (Al-Haj Hemidi et al., 2024), sparse-view CBCT reconstruction (Zha et al., 2022), and image registration (Wolterink et al., 2022; Sideri-Lampretsa et al., 2024).

**Generalizable Neural Fields**    Several approaches have been proposed to generalize NFs from single instances to entire datasets. Earlier methods like SIREN (Sitzmann et al., 2020b), GASP (Dupont et al., 2021b), COIN (Dupont et al., 2021a), or HyperDiffusion (Erkoç et al., 2023) rely on first fitting a set of single-instance NFs, and subsequently using the resulting weights for solving downstream tasks like generation or compression. This is not only expensive, but the unordered nature of the constructed weight spaces also limits the performance of these methods. To bypass first-stage, single-instance NF fitting, auto-decoder frameworks (Park et al., 2019; Stolt-Ansó et al., 2023; Bieder et al., 2024) jointly optimize a shared network and signal-specific latent vectors. Representative methods such as DeepSDF (Park et al., 2019), CINA (Dannecker et al., 2024) and NISF (Stolt-Ansó et al., 2023) follow this strategy, but their naive joint optimization makes fitting new latents at test time slow and computationally expensive. To overcome this limitation, frameworks that *learn-to-learn*, i.e., train the shared network in a way that fitting a new latent at test time requires few steps only, arose. Methods like MetaSDF (Sitzmann et al., 2020a), COIN++ (Dupont et al., 2022a), Functa (Dupont et al., 2022b), SpatialFuncta (Bauer et al., 2023) or LIFT (Kazerouni et al., 2025) use different meta-learning strategies to achieve this goal. A parallel line of research aims to train hypernetworks that generate NF weights, without the need for first-stage single-instance NF fitting (Klocek et al., 2019; Du et al., 2021; Zhuang et al., 2023). They do so by using the auto-decoder framework to learn hypernetworks that map from a low-dimensional, locally linear manifold to the desired NF weights (Du et al., 2021), or by leveraging an explicit field parameterization and score field networks to learn distributions over neural fields in an end-to-end fashion (Zhuang et al., 2023).

While single-instance and generalizable NFs have been studied as a novel *data modality* in the deep learning community (Dupont et al., 2022b; Schürholt et al., 2022b; Ma et al., 2024), their application to medical data, along with the unique challenges and opportunities, remains unexplored.

## Appendix B. Why We Do Not Rely on Patch-Based Representations

Although there is a substantial body of work on generalizable patch-based neural representations, such as COIN++ (Dupont et al., 2022a), SpatialFuncta (Bauer et al., 2023), and LIFT (Kazerouni et al., 2025), the focus of this work is different. We aim to learn a shared representation across signals of varying modalities and dimensions, where each signal is encoded as a single 1D latent vector. While we recognize that the inductive bias of patch-based representations can be advantageous, and while we believe that our findings could also be utilized within these frameworks, such an exploration lies beyond the scope of this work. We argue that a unified representation across diverse signal types is particularly valuable in medicine, where integrating data from multiple modalities remains an active yet unsolved research challenge.

## Appendix C. Implementation Details

In this section, we provide a detailed list of hyperparameters used for the experiments in this paper.

Table 6: The number of layers $K$, the hidden dimension $L$, the batch size $B$, the context selection ratio $\gamma$, $\omega_1$ and $\omega_K$, as well as the representation size $P$.

| Signal Dim. | $K$ | $L$ | $B$ | $\gamma$ | $\omega_1$ | $\omega_K$ | $P$ |
|---|---|---|---|---|---|---|---|
| 1D | 8 | 64 | 64 | 1.00 | 20 | 200 | 64 |
| 2D | 15 | 256 | 24 | 0.25 | 20 | 400 | 2048 |
| 3D | 15 | 256 | 4 | 0.25 | 20 | 300 | 8192 |

## Appendix D. Implementation Details for Comparing Methods

**Functa**   Following the original implementation, we benchmark Functa using a network of depth 15 and a hidden dimension of 512. We set the SIREN frequency parameter to $\omega = 30$ and optimize using 3 inner loop steps during training and inference. Similar to our method, we use a modulation dimension of 2048. As reported in the original paper, we apply SGD with a learning rate of $1 \times 10^{-2}$ for the inner loop updates and Adam with a learning rate of $3 \times 10^{-6}$ for the meta-updates.

**COIN++**   Following the original implementation, we benchmark COIN++ using a network of depth 10 and a hidden dimension of 512. We set the SIREN frequancy parameter to $\omega = 50$ and optimize using 3 inner loop steps during training and 10 inner loop steps during inference. For the unpatched version, we use a modulation dimension of 2048. As reported in the original paper, we apply SGD with a learning rate of $1 \times 10^{-2}$ for the inner loop updates and Adam with a learning rate of $3 \times 10^{-6}$ for the meta-updates.

**COIN++ (patched)**   We additionally benchmark a patched version of COIN++. We apply the same setup as in COIN++, but train on random $32 \times 32$ patches. We therefore scale the modulation dimension from 2048 to 512 to keep the same compression rate. This results in a latent dimension of $2 \times 2 \times 512$. All other hyperparameters are similar to the COIN++ run.

**SpatialFuncta**   Following the hyperparameters reported in the paper, we benchmark SpatialFuncta using a network of depth 12 and a hidden dimension of 256. We set the SIREN frequency parameter to $\omega_0 = 20$ and optimize using 3 inner loop setps during training and inference. We apply a latent dimension of $2 \times 2 \times 512$. As reported in the paper, we apply SGD with a learning rate of $1 \times 10^{-2}$ for the inner loop updates and Adam with a learning rate of $3 \times 10^{-5}$ for the meta-updates.

## Appendix E. Relation Between a SIREN Layer's $\omega$-Parameter and Learning Rate

In this section, we provide a comprehensive mathematical derivation of the relation between a layer's $\omega$-parameter and it's effective learning rate.

**Problem Formulation**   Let's consider two SIREN layers with different frequency parameters $\omega_m \neq \omega_n$. The layers are defined :

$$\mathbf{y}_m = \sin(\omega_m(\mathbf{W}_m\mathbf{x} + \mathbf{b}_m)) \tag{6}$$

$$\mathbf{y}_n = \sin(\omega_n(\mathbf{W}_n\mathbf{x} + \mathbf{b}_n)), \tag{7}$$

where $\mathbf{W}_m, \mathbf{W}_n \in \mathbb{R}^{p \times d}$ are weight matrices, $\mathbf{b}_m, \mathbf{b}_n \in \mathbb{R}^p$ are bias vectors, $\mathbf{x} \in \mathbb{R}^d$ is an input vector, and $\mathbf{y}_m, \mathbf{y}_n \in \mathbb{R}^p$ are output vectors. Our goal is to determine the relationship between the learning rates $\tau_m$ and $\tau_n$ that ensures both layers maintain equivalent outputs throughout training, despite having different frequency parameters.

**Initialization Analysis**   To understand how different $\omega$-values affect a layer's learning rate, we consider the case where both layers are initialized with the same underlying random values but different scaling factors. Specifically, let $\tilde{\mathbf{W}}^{(0)}, \tilde{\mathbf{b}}^{(0)} \sim \mathcal{U}(-\sqrt{6/n}, \sqrt{6/n})$ and define:

$$\mathbf{W}_m^{(0)} = \frac{1}{\omega_m}\tilde{\mathbf{W}}^{(0)}, \quad \mathbf{b}_m^{(0)} = \frac{1}{\omega_m}\tilde{\mathbf{b}}^{(0)} \tag{8}$$

$$\mathbf{W}_n^{(0)} = \frac{1}{\omega_n}\tilde{\mathbf{W}}^{(0)}, \quad \mathbf{b}_n^{(0)} = \frac{1}{\omega_n}\tilde{\mathbf{b}}^{(0)}. \tag{9}$$

This ensures that:

$$\omega_m\mathbf{W}_m^{(0)} = \omega_n\mathbf{W}_n^{(0)} = \tilde{\mathbf{W}}^{(0)} \tag{10}$$

$$\omega_m\mathbf{b}_m^{(0)} = \omega_n\mathbf{b}_n^{(0)} = \tilde{\mathbf{b}}^{(0)}, \tag{11}$$

which means that both layers produce the same output upon initialization:

$$\sin\left(\omega_m(\mathbf{W}_m^{(0)}\mathbf{x} + \mathbf{b}_m^{(0)})\right) = \sin\left(\omega_n(\mathbf{W}_n^{(0)}\mathbf{x} + \mathbf{b}_n^{(0)})\right). \tag{12}$$

**Identifying the Learning Rate Relation**   We now examine the condition that needs to be fulfilled for both layers to maintain equal behavior after gradient descent updates. We therefore require:

$$\omega_m\mathbf{W}_m^{(1)} \stackrel{!}{=} \omega_n\mathbf{W}_n^{(1)}. \tag{13}$$

The gradient descent update steps are defined as:

$$\mathbf{W}_m^{(1)} = \mathbf{W}_m^{(0)} - \tau_m\frac{\partial\mathcal{L}}{\partial\mathbf{W}_m} \tag{14}$$

$$\mathbf{W}_n^{(1)} = \mathbf{W}_n^{(0)} - \tau_n\frac{\partial\mathcal{L}}{\partial\mathbf{W}_n}, \tag{15}$$

where $\tau_m$ and $\tau_n$ are the learning rates for layers $m$ and $n$, respectively. Substituting the update equations (14) and (15) into condition (13), we get:

$$\omega_m\left[\mathbf{W}_m^{(0)} - \tau_m\frac{\partial\mathcal{L}}{\partial\mathbf{W}_m}\right] = \omega_n\left[\mathbf{W}_n^{(0)} - \tau_n\frac{\partial\mathcal{L}}{\partial\mathbf{W}_n}\right]. \tag{16}$$

Using the initialization condition (10), this simplifies to:

$$\omega_m\tau_m\frac{\partial\mathcal{L}}{\partial\mathbf{W}_m} = \omega_n\tau_n\frac{\partial\mathcal{L}}{\partial\mathbf{W}_n}. \tag{17}$$

For a given loss function $\mathcal{L}$, the gradients with respect to the weight matrices are defined as:

$$\frac{\partial \mathcal{L}}{\partial \mathbf{W}_m} = \frac{\partial \mathcal{L}}{\partial \mathbf{y}_m} \frac{\partial \mathbf{y}_m}{\partial \mathbf{W}_m} \tag{18}$$

$$\frac{\partial \mathcal{L}}{\partial \mathbf{W}_n} = \frac{\partial \mathcal{L}}{\partial \mathbf{y}_n} \frac{\partial \mathbf{y}_n}{\partial \mathbf{W}_n}, \tag{19}$$

with:

$$\frac{\partial \mathbf{y}}{\partial \mathbf{W}_m} = \omega_m \mathbf{x}^\top \otimes \mathrm{diag}\Big( \cos\big( \omega_m(\mathbf{W}_m \mathbf{x} + \mathbf{b}_m) \big) \Big) \tag{20}$$

$$\frac{\partial \mathbf{y}}{\partial \mathbf{W}_n} = \omega_n \mathbf{x}^\top \otimes \mathrm{diag}\Big( \cos\big( \omega_n(\mathbf{W}_n \mathbf{x} + \mathbf{b}_n) \big) \Big), \tag{21}$$

where $\otimes$ is the Kronecker product and $\mathrm{diag}(\cdot)$ is the diagonal matrix with the entries of its argument on the diagonal. Substituting the gradient expressions from (20) and (21), Equation (17) becomes:

$$\begin{aligned}
&\omega_m \tau_m \left( \frac{\partial \mathcal{L}}{\partial \mathbf{y}_m} \omega_m \mathbf{x}^\top \otimes \mathrm{diag}\Big( \cos\big( \omega_m(\mathbf{W}_m \mathbf{x} + \mathbf{b}_m) \big) \Big) \right) \\
&= \omega_n \tau_n \left( \frac{\partial \mathcal{L}}{\partial \mathbf{y}_n} \omega_n \mathbf{x}^\top \otimes \mathrm{diag}\Big( \cos\big( \omega_n(\mathbf{W}_n \mathbf{x} + \mathbf{b}_n) \big) \Big) \right)
\end{aligned} \tag{22}$$

Due to the initialization conditions (10) and (11), the arguments of the cosine functions are the same:

$$\omega_m(\mathbf{W}_m^{(0)}\mathbf{x} + \mathbf{b}_m^{(0)}) = \omega_n(\mathbf{W}_n^{(0)}\mathbf{x} + \mathbf{b}_m^{(0)}). \tag{23}$$

The same holds true for the loss gradients with respect to the model outputs - considering the similar output from Equation (12):

$$\frac{\partial \mathcal{L}}{\partial \mathbf{y}_m} = \frac{\partial \mathcal{L}}{\partial \mathbf{y}_n} \tag{24}$$

Equation (22) therefore reduces to:

$$\omega_m^2 \tau_m = \omega_n^2 \tau_n, \tag{25}$$

which can be reformulated as:

$$\frac{\tau_n}{\tau_m} = \left( \frac{\omega_m}{\omega_n} \right)^2 \tag{26}$$

## Appendix F. Additional Details on the Meta-Learning Approach

This section provides further insights into our meta-learning framework summarized in Algorithm 1. We start with the **inner-loop optimization** process in which the signal-specific parameter vectors are initialized as a zero-vector:

$$\phi_0^{(i)} := \mathbf{0}, \tag{27}$$

and updated for $g = 0, ..., G - 1$ update steps using stochastic gradient descent (SGD):

$$\phi_{g+1}^{(i)} = \phi_g^{(i)} - \alpha \nabla_\phi \mathcal{L}_{\texttt{MSE}}(\phi_g^{(i)}, \theta; \mathcal{C}^{(i)}). \tag{28}$$

After performing $G$ inner-loop update steps, the **meta-objective**, used for updating the shared network parameters $\theta$, is defined as:

$$\mathcal{L}_{\text{meta}} = \frac{1}{B} \sum_{i=1}^{B} \mathcal{L}_{\texttt{MSE}}(\phi_G^{(i)}, \theta; \mathcal{C}^{(i)}), \tag{29}$$

for a batch with $B$ signals, where $\phi_G^{(i)}$ denotes the adapted signal-specific parameters, which are themselves functions of $\theta$. The **gradient of the meta-objective** with respect to $\theta$ is therefore defined as:

$$\nabla_\theta \mathcal{L}_{\text{meta}}(\theta) = \frac{1}{B} \sum_{i=1}^{B} \nabla_\theta \mathcal{L}_{\texttt{MSE}}(\phi_G^{(i)}, \theta; \mathcal{C}^{(i)})$$

$$= \frac{1}{B} \sum_{i=1}^{B} \left[ \underbrace{\nabla_\theta \mathcal{L}_{\texttt{MSE}}(\phi_G^{(i)}, \theta; \mathcal{C}^{(i)})}_{\text{direct effect}} + \underbrace{\left( \frac{\partial \phi_G^{(i)}}{\partial \theta} \right)^{\top} \nabla_\phi \mathcal{L}_{\texttt{MSE}}(\phi_G^{(i)}, \theta; \mathcal{C}^{(i)})}_{\text{indirect effect via } \phi_G^{(i)}} \right]. \tag{30}$$

Computing the meta-gradient requires differentiating through the inner-loop optimization process. Each signal-specific parameter vector $\phi_G^{(i)}$ is the result of $G$ gradient descent steps that depend on the current shared parameters $\theta$. Therefore, when taking the derivative of the meta-objective with respect to $\theta$, we must account for both:

- **Direct effect:** the explicit influence of $\theta$ on the loss $\mathcal{L}_{\texttt{MSE}}$ given the adapted parameters $\phi_G^{(i)}$.

- **Indirect effect:** the influence of $\theta$ on the loss through its effect on the inner-loop parameters $\phi_G^{(i)}$.

While taking the direct effect into account is straightforward, the indirect effect is captured by the term:

$$\left( \frac{\partial \phi_G^{(i)}}{\partial \theta} \right)^{\top} \nabla_\phi \mathcal{L}_{\texttt{MSE}}(\phi_G^{(i)}, \theta; \mathcal{C}^{(i)}), \tag{31}$$

which involves second-order derivatives of the inner-loop loss. To compute this term, we differentiate through the inner-loop recursion, which yields the following recursive formula for the Jacobian of the adapted parameters with respect to $\theta$:

$$\frac{\partial \phi_{g+1}^{(i)}}{\partial \theta} = \frac{\partial \phi_g^{(i)}}{\partial \theta} - \alpha \left( \frac{\partial^2 \mathcal{L}_{\texttt{MSE}}(\phi_g^{(i)}, \theta; \mathcal{C}^{(i)})}{\partial \theta \, \partial \phi} + \frac{\partial^2 \mathcal{L}_{\texttt{MSE}}(\phi_g^{(i)}, \theta; \mathcal{C}^{(i)})}{\partial \phi^2} \frac{\partial \phi_g^{(i)}}{\partial \theta} \right). \tag{32}$$

This recursion explicitly shows how the inner-loop updates propagate the influence of $\theta$ to the adapted parameters, and why second-order terms (Hessian-vector products) are required to compute the full meta-gradient. After computing the required gradient, we take a single **meta-update** of the shared parameters $\theta$ using AdamW with learning rate $\beta$:

$$\theta \leftarrow \texttt{AdamW}(\theta, \nabla_\theta \mathcal{L}_{\text{meta}}, \beta). \tag{33}$$

---

**Algorithm 1:** Meta-Learning with Context Reduction During Inner-Loop Optimization

---

**Input:** Dataset $\mathcal{D} = \{s_1, s_2, \ldots, s_N\}$, inner-loop steps $G$, inner-loop learning rate $\alpha$, meta learning rate $\beta$, selection ratio $\gamma$, batch size $B$

**Output:** Trained meta-parameters $\theta$

**while** *not converged* **do**

    $\{s_i\}_{i=1}^{B} \sim \mathcal{D}$                 // Sample batch of $B$ signals

    $\mathcal{L}_{\text{meta}} \leftarrow 0$                 // Reset meta-loss

    **for** *each signal $s_i$ in batch* **do**

        $\phi_0^{(i)} \leftarrow \mathbf{0}$              // Initialize $\phi^{(i)}$

        **for** $g = 0$ **to** $G - 1$ **do**

            $\mathcal{C}_{\text{red}}^{(i)} \sim \texttt{Sample}\big(\mathcal{C}^{(i)}, \gamma|\mathcal{C}^{(i)}|\big)$     // Sample reduced context set

            $\phi_{g+1}^{(i)} \leftarrow \phi_g^{(i)} - \alpha \nabla_\phi \mathcal{L}_{\texttt{MSE}}\big(\phi_g^{(i)}, \theta; \mathcal{C}_{\text{red}}^{(i)}\big)$     // Inner-loop update

        **end**

        $\mathcal{L}_{\text{meta}} \leftarrow \mathcal{L}_{\text{meta}} + \mathcal{L}_{\texttt{MSE}}\big(\phi_G^{(i)}, \theta; \mathcal{C}^{(i)}\big)$     // Accumulate meta-loss

    **end**

    $\mathcal{L}_{\text{meta}} \leftarrow \frac{1}{B}\mathcal{L}_{\text{meta}}$           // Average over batch

    $\theta \leftarrow \texttt{AdamW}\big(\theta, \nabla_\theta \mathcal{L}_{\text{meta}}, \beta\big)$         // Meta-update

    Update $\beta$ according to learning rate schedule     // Learning rate update

**end**

**return** $\theta$

---

Table 7: A list of all released sub-datasets with their original data source, their modality, the task we provide labels for, the number of samples, as well as the training/validation/test split. All datasets were adapted from MedMNIST (Yang et al., 2021, 2023).

| Dataset | Modality | Task | Samples | Train / Val / Test | License |
|---------|----------|------|---------|--------------------|---------|
| ChestNF | Chest X-Ray | Multi-Label (14) Binary-Class (2) Classification | 112 120 | 78 468 / 11 219 / 22 433 | CC BY 4.0 |
| PathNF | Colon Pathology | Multi-Class (9) Classification | 107 180 | 89 996 / 10 005 / 7180 | CC BY 4.0 |
| DermaNF | Dermatoscope | Multi-Class (7) Classification | 10 015 | 7007 / 1003 / 2005 | CC BY-NC 4.0 |
| OctNF | Retinal OCT | Multi-Class (4) Classification | 109 309 | 97 477 / 10 832 / 1000 | CC BY 4.0 |
| PneumoniaNF | Chest X-Ray | Binary-Class (2) Classification | 5856 | 4708 / 524 / 624 | CC BY 4.0 |
| RetinaNF | Fundus Camera | Ordinal Regression (5) | 1600 | 1080 / 120 / 400 | CC BY 4.0 |
| TissueNF | Kidney Cortex Microscope | Multi-Class (8) Classification | 236 386 | 165 466 / 23 640 / 47 280 | CC BY 4.0 |

## Appendix G. Details on MedNF

In this section, we will provide further information on our MedNF datasets. All datasets, listed in Table 7, are adapted from MedMNIST (Yang et al., 2021, 2023). We meta-learned every model on the respective training set for 250 k iterations, using the setup described in Section 3. All datasets were derived from images with a resolution of $64 \times 64$. We also aim to release models trained on higher and mixed resolutions in the future.

**ChestNF** is build upon the NIH-ChestXray14 (Wang et al., 2017) dataset and contains 112 120 frontal-view chest X-Ray images. It also provides binary disease-labels for the following diseases: (0) atelectasis, (1) cardiomegaly, (2) effusion, (3) infiltration, (4) mass, (5) nodule, (6) pneumonia, (7) pneumothorax, (8) consolidation, (9) edema, (10) emphysema, (11) fibrosis, (12) pleural, (13) hernia.

**PathNF** is build upon the NCT-CRC-HE-100K (Kather et al., 2019) dataset and contains 107 180 non-overlapping image patches of hematoxylin and eosin-stained colorectal cancer histology slides. It also provides a class label for one of nine tissues: (0) adipose, (1) background, (2) debris, (3) lymphocytes, (4) mucus, (5) smooth muscle, (6) normal colon mucosa, (7) cancer-associated stroma, (8) colorectal adenocarcinoma epithelium.

**DermaNF** is build upon the HAM10000 (Codella et al., 2019; Tschandl et al., 2018) dataset and contains 10 015 dermatoscopic images of common pigmented skin lesions. It also provides a class label for one of the seven following diseases: (0) actinic keratoses and intraepithelial carcinoma, (1) basal cell carcinoma, (2) benign keratosis-like lesions, (3) dermatofibroma, (4) melanoma, (5) melanocytic nevi, (6) vascular lesions.

**OctNF** is build upon a dataset from (Kermany et al., 2018) and contains 109 309 optical coherence tomography (OCT) images for retinal diseases. It also provides a class label for one of the four following diseases: (0) choroidal neovascularization, (1) diabetic macular edema, (2) drusen, (3) normal.

**PneumoniaNF** is build upon a dataset from (Kermany et al., 2018) and contains 5856 pediatric chest X-Ray scans. It contains the following binary label: (0) normal, (1) pneumonia.

**RetinaNF** is build upon the DeepDRiD (Liu et al., 2022) dataset and contains 1600 retina fundus images. It provides labels for a 5-level grading of diabetic retinopathy severity.

**TissueNF** is build upon the BBBC051 (Ljosa et al., 2012) dataset from the Broad Bioimag-

ing Benchmark Collection and contains 236 386 human kidney cortex cell images. It provides labels for one of the eight following cell types: (0) collecting duct, connecting tubule, (1) distal convoluted tubule, (2) glomerular endothelial cells, (3) interstitial endothelial cells, (4) leukocytes, (5) podocytes, (6) proximal tubule segments, (7) thick ascending limb.

## Appendix H. Negative Results

While the following explorations did not lead to promising results, we include them for completeness and transparency. We believe that reporting negative results helps to clarify the scope of our contributions, reduce redundant future efforts, and provide insight into design choices that, while theoretically appealing, did not prove effective in practice.

As our method constructs a representation space, i.e., the space in which the signal-specific parameter vectors $\phi$ reside, we explored **incorporating a supervised contrastive loss** (Khosla et al., 2020) into our meta-learning objective. While this led to modest improvements in a simple binary classification task, we observed no improvement in more challenging multi-class classification settings. We assume that this is due to the lack of a learned encoder, usually available in representation learning settings. Consequently, we did not pursue this approach further.

We also experimented with **quantized representations**, where each entry is drawn from a learned codebook. Inspired by ideas from vector-quantized autoencoders (Van Den Oord et al., 2017), we hypothesized that this could improve performance by simplifying the decoding task for the shared network. However, we observed no such benefits and therefore retained continuous representations.

Based on the observation that shallow layers primarily capture relatively simple, low-frequency features while deeper layers encode more complex, high-frequency details, we hypothesized that a **pyramid-like network structure** might improve performance (Chen et al., 2023). In this design, shallow layers would contain fewer neurons, reflecting the lower complexity of low-frequency features, while deeper layers would be allocated more neurons to better model the harder high-frequency details. However, this approach did not yield substantial improvements.

We additionally explored **first-order approximations** of our meta-learning framework, namely first-order MAML with and without our proposed $\omega$-schedule. Similar to (Dupont et al., 2022a,b), we experienced training instability and a largely reduced reconstruction performance of $> 10\,\text{dB}$ on `Chest X-ay` $(64 \times 64)$ across both settings, ultimately retaining second-order optimization.

Lastly, instead of randomly sampling a subset of pixels in our context reduction approach, we experimented with **gradient-based context pruning**, as proposed in (Tack et al., 2023). However, this method did not outperform random subsampling and, in some cases, led to unstable and collapsing training. We therefore decided to continue using random subsampling.

## Appendix I. Additional Comparisons

In this section, we present additional comparisons to Functa (Dupont et al., 2022b). In Figure 5, we plot the evolution of PSNR, MSE, SSIM, and LPIPS on the validation set over

the course of training. Our method not only achieves superior scores in significantly less time, it also exhibits more stable training dynamics.

We also provide additional qualitative comparisons to Functa, COIN++, COIN++ (patched), and SpatialFuncta, along with absolute difference maps, in Figure 6. Our model not only surpasses all methods operating at the same spatial resolution, but also outperforms existing patch-based approaches. We believe that integrating the insights presented in this paper into patch-based methods could further enhance their performance and support their continued scaling.

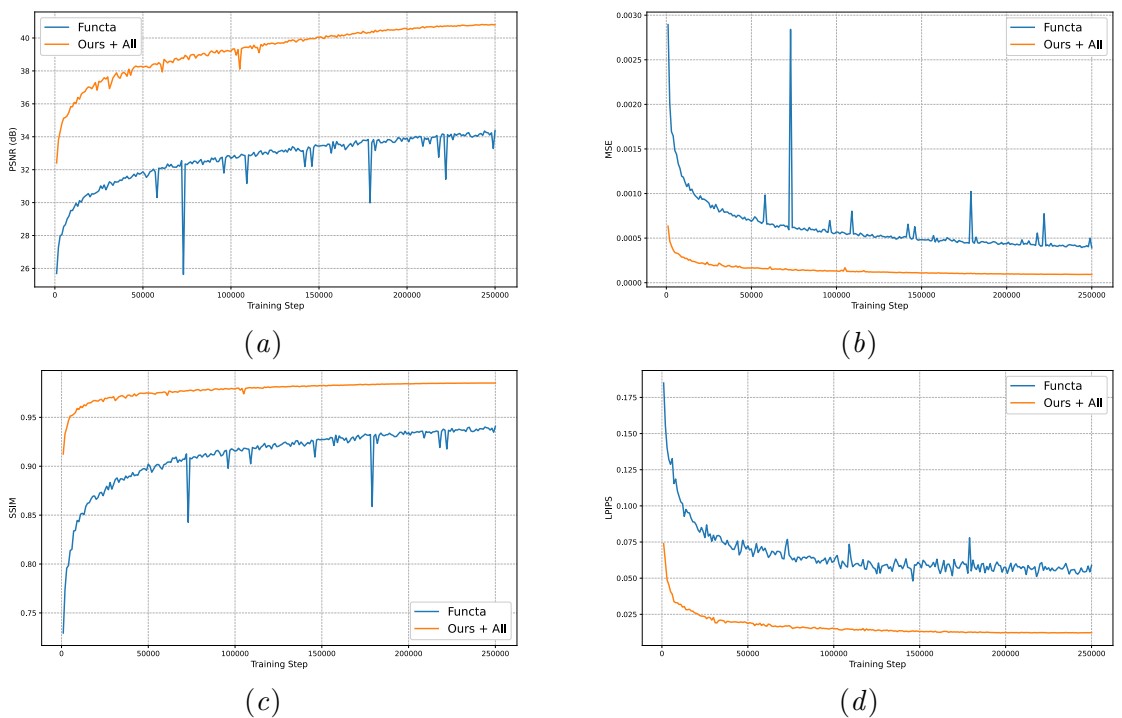

Figure 5: Development of (a) PSNR, (b) MSE, (c) SSIM, and (d) LPIPS on the validation set throughout a training run (250 k iterations).

## Appendix J. Additional Qualitative Results

In this section, we provide further qualitative results for the experiments in Section 3. The results are shown in Figure 7 - Figure 19.

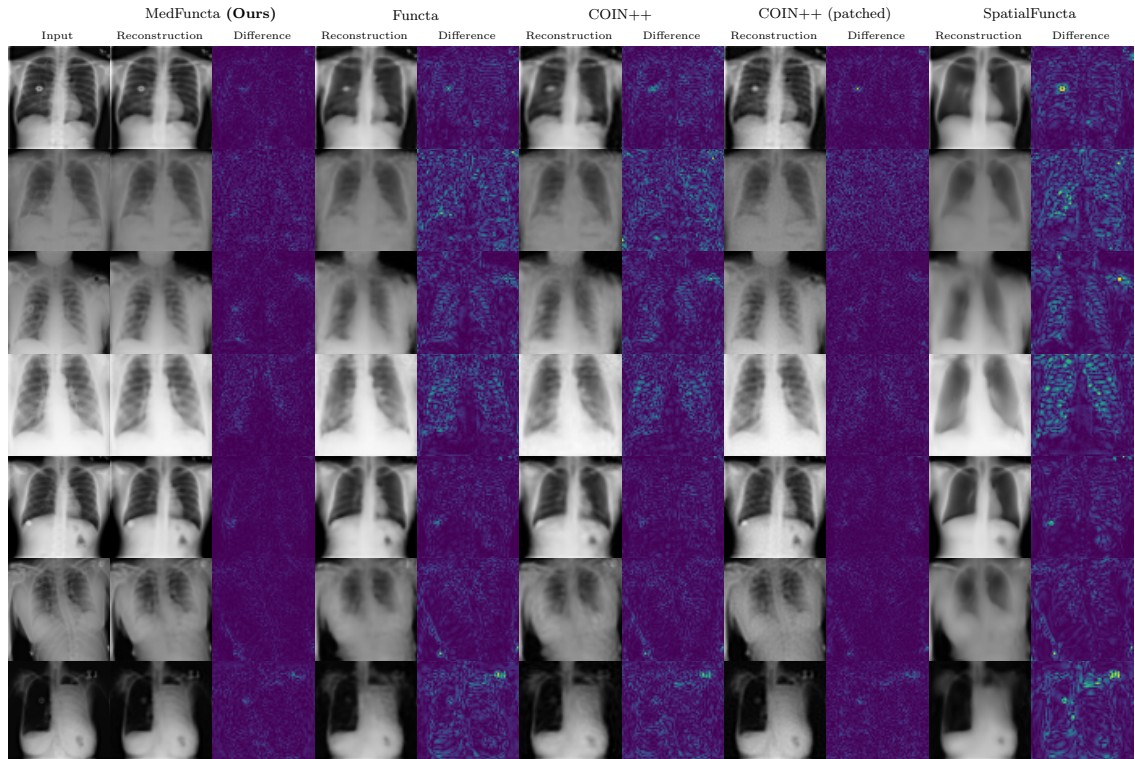

Figure 6: Qualitative comparison between our proposed approach MedFuncta, Functa, COIN++, COIN++ (patched), and SpatialFuncta with absolute difference maps. Best viewed zoomed.

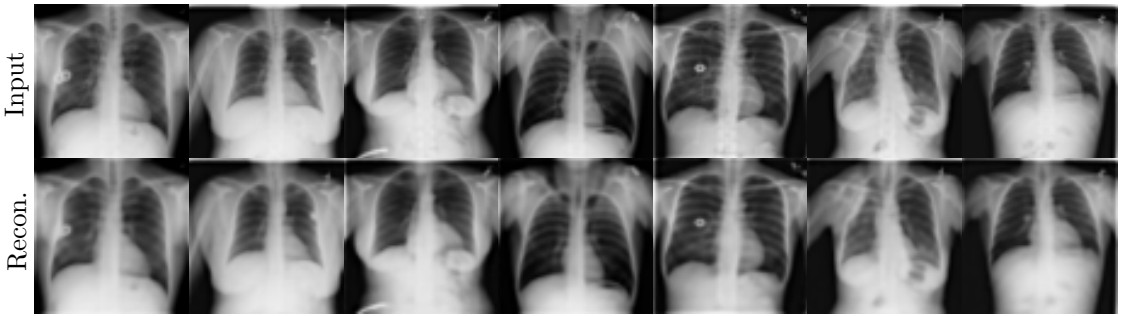

Figure 7: Additional qualitative results on `Chest X-ray` images ($64 \times 64$).

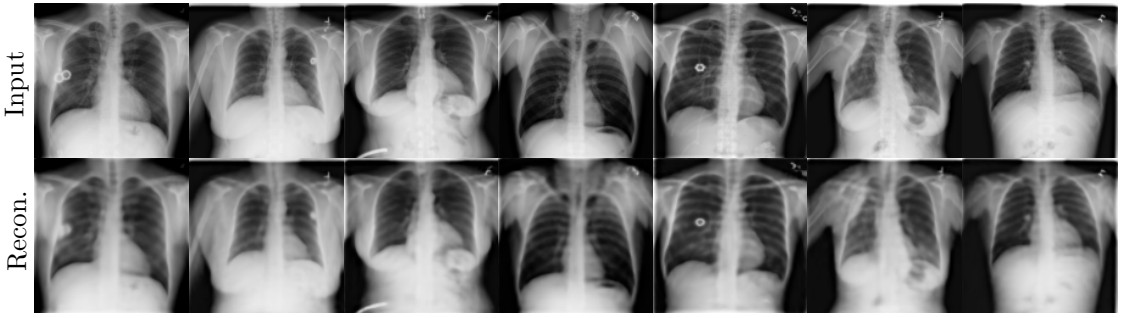

Figure 8: Additional qualitative results on `Chest X-Ray` images ($128 \times 128$).

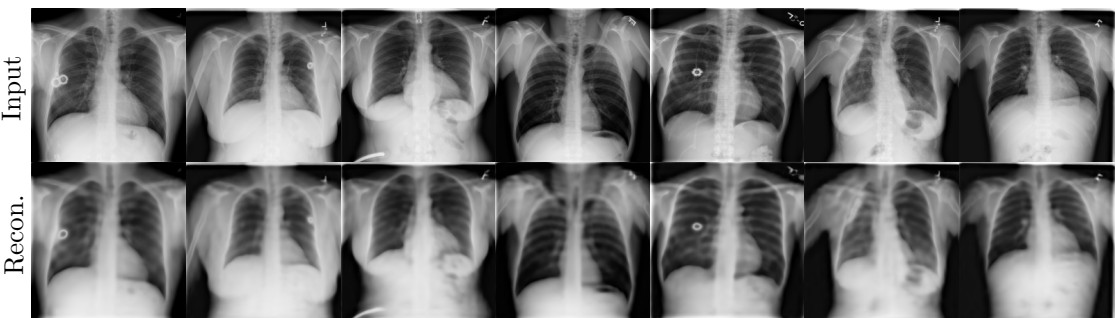

Figure 9: Additional qualitative results on `Chest X-Ray` images ($224 \times 224$).

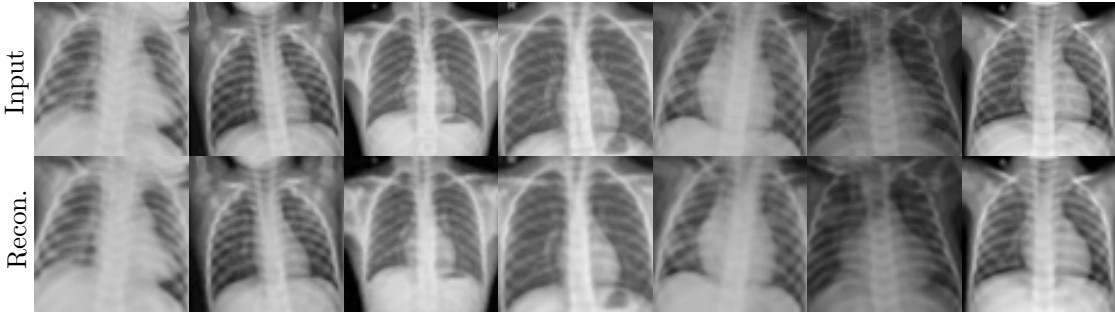

Figure 10: Additional qualitative results on `Pneumonia Chest X-Ray` images ($64 \times 64$).

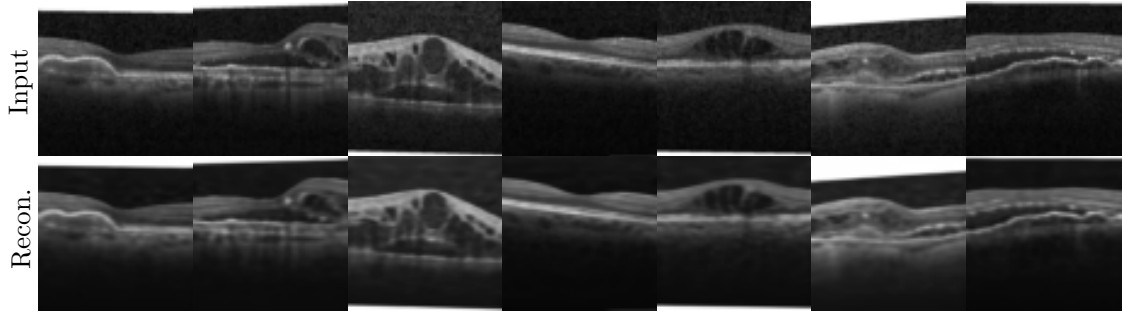

Figure 11: Additional qualitative results on `Retinal OCT` images ($64 \times 64$).

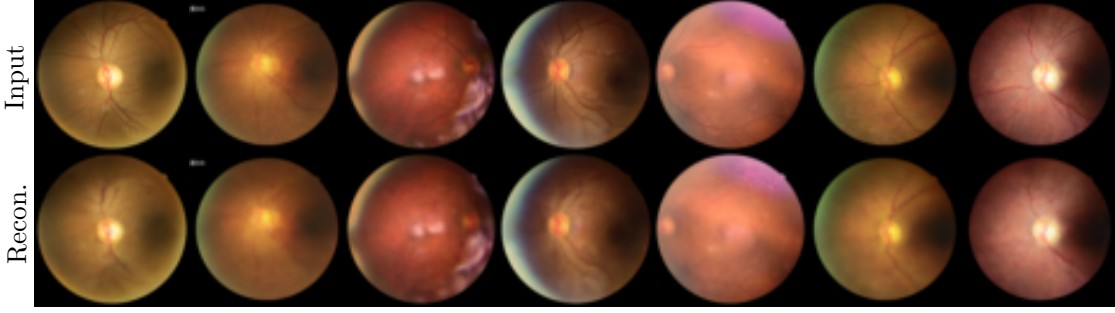

Figure 12: Additional qualitative results on `Fundus Camera` images ($64 \times 64$).

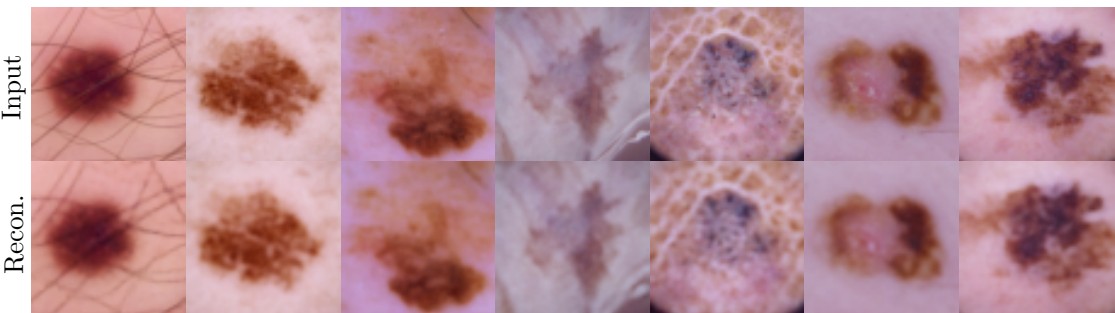

Figure 13: Additional qualitative results on `Dermatoscope` images ($64 \times 64$).

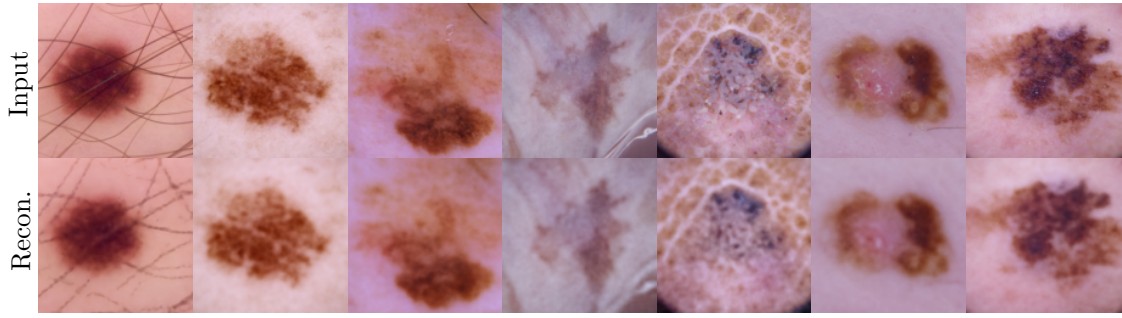

Figure 14: Additional qualitative results on `Dermatoscope` images ($128 \times 128$).

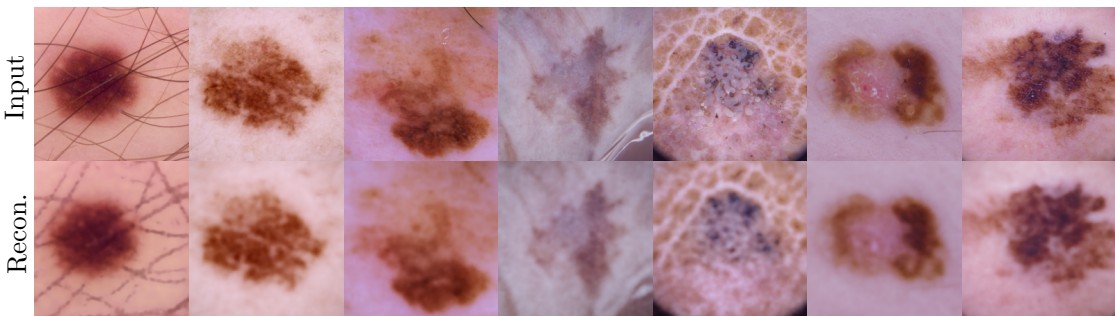

Figure 15: Additional qualitative results on `Dermatoscope` images ($224 \times 224$).

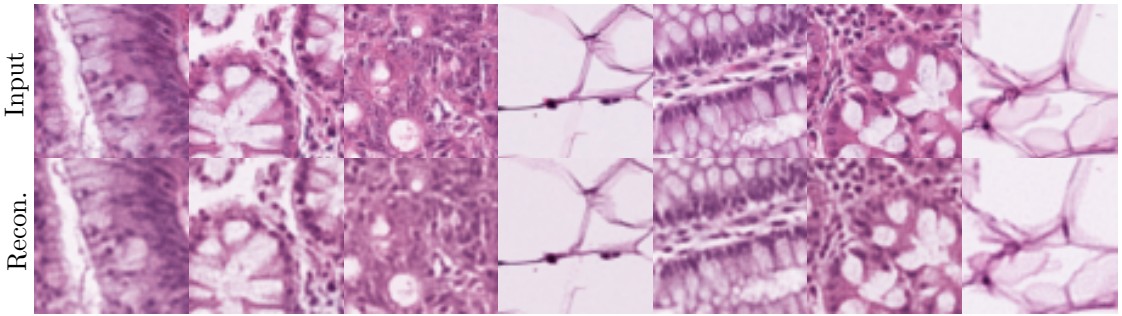

Figure 16: Additional qualitative results on `Histopathology` images ($64 \times 64$).

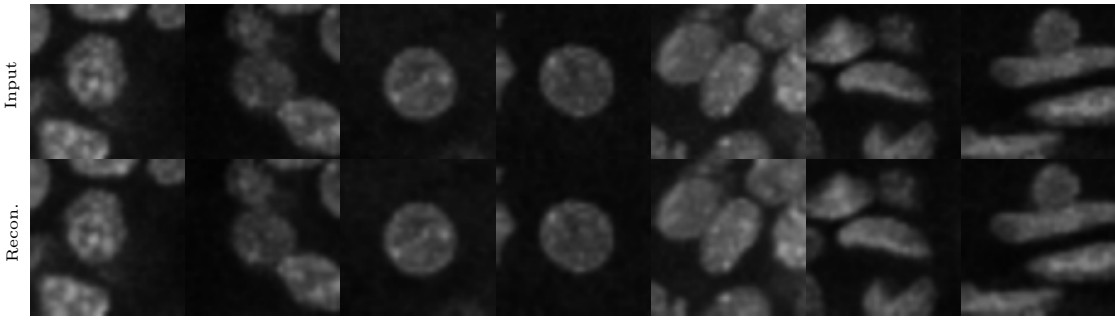

Figure 17: Additional qualitative results on `Cell Microscopy` images ($64 \times 64$).

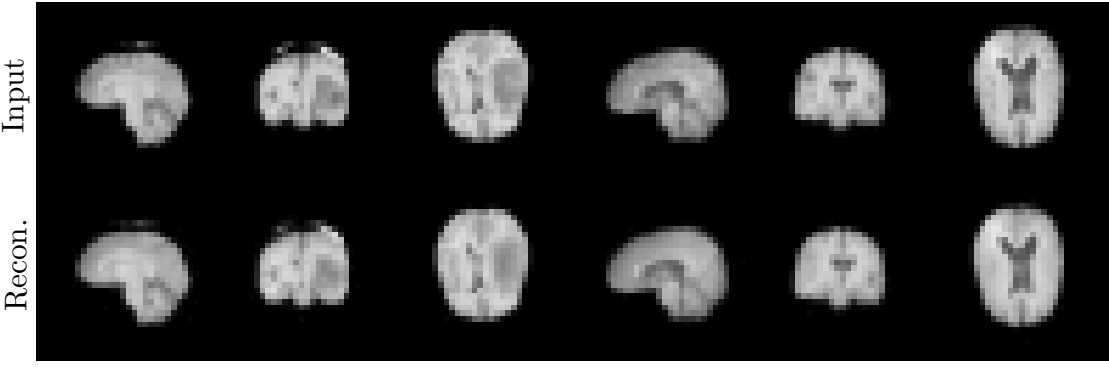

Figure 18: Qualitative results on `Brain MRI` images ($32 \times 32 \times 32$). We display the middle slices along all spatial dimensions.

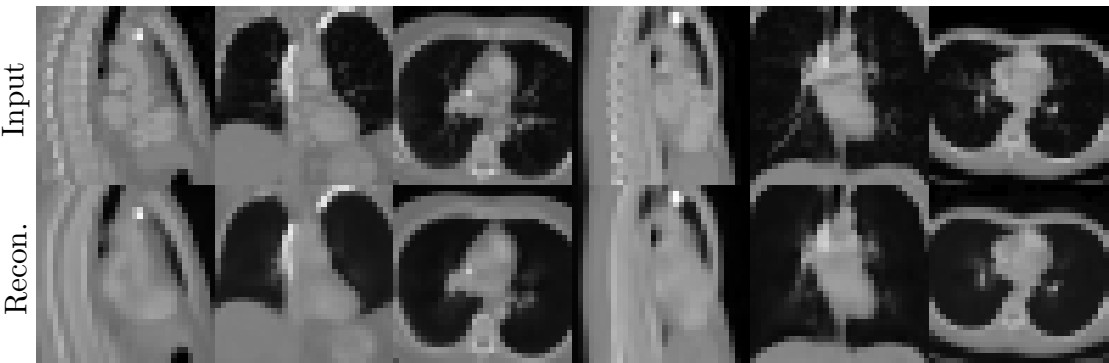

Figure 19: Qualitative results on Lung CT images ($32 \times 32 \times 32$). We display the middle slices along all spatial dimensions.

