# OpenReview forum: "MedFuncta: A Unified Framework for Learning Efficient Medical Neural Fields"
_MIDL.io/2026/Conference — MIDL 2026 Poster_

### Official Review · Reviewer_pR1Y · 2025-12-29

**Confidence:** 5
**Preliminary Rating:** 3
**Final Rating:** 4

**Summary:**

This paper introduces MedFuncta, a framework for learning generalizable neural fields across medical imaging datasets. Building on Functa, the method meta-learns shared network parameters θ while representing individual signals through signal-specific modulation vectors φ. The key contributions are: (1) a layer-wise ω-schedule for SIREN activations that connects frequency parameters to effective learning rates, (2) a context reduction strategy that reduces memory overhead during second-order meta-learning by using sparse supervision, and (3) extensive evaluation across diverse medical datasets along with release of MedNF, a large-scale dataset containing >500k latent vectors.

**Strengths:**

S1: Comprehensive 2D experimental evaluation. The paper evaluates on multiple diverse medical datasets spanning different imaging modalities (X-ray, OCT, fundoscopy, dermatoscopy, histopathology, microscopy). This breadth convincingly demonstrates the framework's flexibility for 2D medical imaging.

S2: Practical efficiency gains from context reduction. The context reduction strategy achieves ~70% memory reduction and >50% training time reduction with minimal performance degradation (<1 dB PSNR, Table 4). This is meaningful for scaling to high-dimensional medical data without requiring first-order approximations.

S3: Significant open science contribution. The release of code, model weights, and the MedNF dataset (>500k latent vectors across seven 2D datasets) will benefit the research community and enable reproducibility and future research on neural field representations.

S4: Strong reconstruction quality improvements. The method achieves high PSNR/SSIM across 2D datasets and the full method outperforms Functa by ~6.4 dB on Chest X-ray while using fewer parameters (7.7M vs 17.1M).

**Weaknesses:**

W1: The "unified" framework uses separate models per dimensionality. Table 6 reveals substantially different network configurations for 1D, 2D, and 3D signals (e.g., K=8 vs 15 layers, L=64 vs 256 hidden dim, P=64 vs 8192 latent size). The claim of a "unified framework" can read misleading—there is no single model handling different dimensions, only a shared methodology. The paper should clarify what "unified representation" means in practice.

W2: The classification experiments (Table 3) compare against ResNet50 and EfficientNet-B0 trained from scratch for only 50 epochs with fixed learning rate. This underestimates baseline performance: no pretrained ImageNet weights (standard for small medical datasets), no learning rate scheduling, and no data augmentation are mentioned. It makes sense to compare to models trained in similar settings, but additional benchmarks should be provided that serve as upper bounds. The claim of "outperforming ResNet50" is not well-supported.

W3: The paper motivates context reduction as a solution to second-order memory overhead, citing prior work claiming first-order methods fail. However, no direct comparison is provided for the proposed architecture. This is critical because first-order methods would more directly solve the memory problem, the ω-schedule might stabilize first-order training, and the success with γ=0.25 suggests the problem may be well-conditioned enough for first-order methods.

W4: Despite claiming to handle "1D time series to 3D volumetric data," the 3D evaluation is notably incomplete: no visualizations (Figures 7-17 cover all 2D datasets but zero 3D), no downstream tasks (classification only on 2D), exclusion from MedNF release (only 2D latents), and notably lower reconstruction quality (Lung CT: 28.3 dB vs 40+ dB for most 2D).

W5: The proof in Appendix E has important caveats: (a) single-step analysis that may not hold over 250k iterations, (b) the derived equivalence condition may not be the optimization goal, and (c) potential performance ceiling from the ω-learning rate trade-off is not discussed.

**Detailed Comments:**

**On the ω-schedule theory (Section 2, Appendix E):**

The derivation establishing τₙ/τₘ = (ωₘ/ωₙ)² is elegant but raises questions. The proof shows that gradient magnitude scales as ω while initialization scales as 1/ω, so relative gradient magnitude scales as ω². This seems to contradict the "lower effective learning rate for high ω" interpretation when considering the full optimization picture. The paper would benefit from:
- Extending analysis beyond single-step dynamics
- Providing empirical validation through per-layer convergence curves
- Discussing whether deep layers (high ω) fully converge or represent a bottleneck

**On architectural choices:**

The choice of modulated SIRENs over alternatives lacks justification:
- Hash encodings (Instant NGP) offer faster convergence with comparable quality
- Hybrid approaches (hash features + learned decoder) could combine benefits
- The ω-schedule insights may not transfer to other architectures, limiting broader impact

A comparison against Instant NGP, even in single-instance mode, would contextualize whether meta-learning overhead is worthwhile.

**On 3D evaluation:**

The Lung CT result (28.3 dB PSNR) is substantially below all 2D results. Without visualizations, we cannot assess: (a) whether this represents acceptable clinical quality, (b) slice-wise consistency across the volume, or (c) types of artifacts present. For a paper claiming dimensionality-agnostic representation, this would be a requirement.

**On classification experiments:**

Beyond the baseline fairness issue (training from scratch without pretrained weights, fixed LR, only 50 epochs), the experiment design has further limitations:
- Only two datasets evaluated (Pneumonia: binary, Dermatoscope: 7-class)
- Both are 2D—no 3D classification despite available labels (BraTS tumor grading, LIDC-IDRI nodule malignancy)

**On the MedNF dataset:**

While valuable, the dataset release excludes: all 1D data (ECG) and all 3D data (Brain MRI, Lung CT). This limits the community's ability to explore the full scope of claimed capabilities.

**Minor comments:**
- Latent space visualizations (t-SNE/UMAP of φ vectors colored by class) would demonstrate whether meaningful structure emerges in the learned representation
- Learning curves showing convergence behavior across datasets would help assess whether different datasets require different training durations
- The negative results in Appendix H (supervised contrastive loss, quantization, pyramid architectures) are valuable for transparency; consider briefly mentioning in main text
- Compression ratio compared to storing raw images is not discussed, despite connections to COIN/COIN++
- Notation inconsistency between φ and ϕ throughout the paper
- Page 9 uses "w₁" instead of "ω₁" in the grid search description

**Justification Of Final Rating:**

The authors have addressed most of my concerns, with some minor questions regarding baseline comparisons remaining. Nonetheless, it is my opinion that the general idea of standardizing the learning of representation in INRs is very valuable, which the authors make a big contribution towards. I have therefore raised my score.

**Justification Of The Preliminary Rating:**

This paper presents genuinely interesting ideas with practical value, but significant gaps between claims and evidence prevent me from recommending acceptance in its current form. On the positive side, the ω-schedule contribution is novel and theoretically motivated—the connection between SIREN's frequency parameter and effective layer-wise learning rates (τ ∝ 1/ω²) offers a fresh perspective, with clear empirical improvements demonstrated in Table 5 and Figure 4. The context reduction strategy achieves ~70% memory reduction with minimal performance loss, which is practically valuable for high-dimensional medical data. The breadth of 2D experiments is commendable, and the MedNF dataset release (>500k latent vectors) represents a meaningful contribution to the community. However, several issues hold the paper back. Some critical baselines are missing—no first-order optimization comparison (FOMAML, Reptile) weakens the motivation for second-order meta-learning, and no comparison against hash encodings (Instant NGP) leaves the SIREN architecture choice unjustified. The paper claims a "unified framework" for 1D/2D/3D signals, but 3D evaluation consists only of two rows in Table 1 with no visualizations, no downstream tasks, and exclusion from the released dataset. The Lung CT result (28.3 dB) is notably weaker than 2D results without explanation.

The paper introduces valuable ideas with demonstrated benefits, but incomplete experimental validation places it at the borderline, and a strong rebuttal addressing these critical questions would shift my assessment toward acceptance.

**Questions To Address In The Rebuttal:**

The following questions focus on points that could change my assessment of the paper:

1. Can you provide a first-order optimization comparison (FOMAML, Reptile) with and without the ω-schedule? Even if performance drops, quantifying the trade-off between memory savings and reconstruction quality is essential to justify the second-order approach. Does the ω-schedule stabilize first-order methods compared to constant ω?

2. Can you provide reconstruction visualizations for Brain MRI and Lung CT, including multiple slices to assess volumetric consistency? The Lung CT PSNR (28.3 dB) is notably lower than 2D results—what explains this gap (data complexity, resolution constraints, suboptimal hyperparameters)?

4. Why are hash encoding methods (Instant NGP) not compared against? What prevents applying multi-resolution hash encodings in the meta-learning framework? How does your test-time adaptation speed compare to simply fitting a fast hash-based network per signal independently?

5. Can you report published MedMNIST benchmark results alongside your classification comparison in Table 3, or include properly tuned baselines with pretrained ImageNet weights, learning rate scheduling, and data augmentation?

6. Why are 3D datasets (Brain MRI, Lung CT) and 1D datasets (ECG) excluded from the MedNF release? Is this a technical limitation or planned for future work?

7. The theoretical analysis in Appendix E considers only a single gradient step. Do you have evidence (e.g., per-layer loss curves, gradient statistics tracked over training) that the staged learning dynamics persist throughout the full 250k iterations?

8. How sensitive is the method to ω₁ and ωK choices in practice? Figure 4 suggests dataset-dependent optima. Is per-dataset hyperparameter tuning required, and if so, how expensive is this search?

9. Can φ vectors from different datasets (e.g., ChestNF and DermaNF) be meaningfully compared or interpolated, given they condition the same shared architecture? This would strengthen the "unified representation" claim.

---

> ### Author Response · Authors · 2026-01-22
> **Response to Reviewer pR1Y - Part 1**
>
> We would like to thank the Reviewer for the comprehensive feedback and valuable comments on our work. We will focus on your specific questions and will make sure to include your comments into the revised version of the paper.
>
> **(1) First-order approaches:** Great point! Following your feedback, we ran two additional experiments (FOMAML with and FOMAML without our proposed $\omega$-schedule) to be able to properly address this question. Both variants show training instability and result in largely reduced performance (reconstruction < 31 dB PSNR on the Chest X-Ray $64 \times 64$ dataset, which is more than 10 dB difference). This is in line with what other researchers have reported [1, 2] and justifies why second-order optimization, while being expensive, is still worth doing. We have added these findings to the "Negative Results" section of the revised manuscript to (a) justify our decision and (b) share the results with other readers of the paper. Please also check our response to Reviewer QtRf which includes some ideas from our ongoing work to make the method (even) more efficient and scalable.
>
> [1] Dupont, E, et al. "*From data to functa: Your data point is a function and you can treat it like one.*" ICML 2022.
>
> [2] Dupont, E, et al. "*COIN++: Neural Compression Across Modalities.*" TMLR 2022.
>
> **(2) Qualitative results for 3D datasets:** We added qualitative results of 3D reconstructions to the Appendix of the revised version. The performance gap observed for 3D datasets, as mentioned in the paper, can be explained by the fact that the 3D models ran under different conditions due to hardware constraints. As previously mentioned, we are actively working on this, despite having made significant improvements over e.g. the Functa baseline.
>
> **(3) Why we don't compare to InstantNGP:** Hash-grid methods like InstantNGP store information in spatially-localized feature tables, where each entry corresponds to specific spatial locations. This design fundamentally conflicts with the proposed meta-learning objective. The learned hash table entries encode scene-specific spatial patterns that by-design don't generalize across signals with different content or geometry. In contrast, MLP weights learn functional mappings that can capture transferable priors about signal structure, smoothness, and local correlations (properties that are shared across signals in a domain). Meta-learning the MLP-decoder of these methods would be possible, but as the hash-grid carries most of the information wouldn't really make sense.
>
> **(4) Report other MedMNIST results:** We deliberately refrain from including comparisons with other published results, as we can not ensure that these models were trained under comparable conditions, leading to a potentially biased evaluation. We would like to clarify that the goal of this experiment was not to train state-of-the-art classification models, but rather to demonstrate that our proposed representation actually captures meaningful and usable information about the underlying signals. This also motivates the applied training setting that doesn't apply advanced augmentation strategies, learning rate schedulers, etc. We want to isolate the performance from these effects and study how much information is contained in our representation and how it can be extracted. We agree that exploring more advanced training paradigms, especially augmentation strategies for our signal representation, is a relevant direction for future work.
>
> **(5) Why we exclude some datasets from our dataset release:** We are working on a release of the 1D & 3D datasets (possible due to CC license). As we want to ensure high-quality datasets with useful labels, this will still take some time. It is not a technical limitation.
>
> **(6) Clarifications on Appendix E:** The role of Appendix E is to provide a mechanistic interpretation of why varying $\omega$ across layers affects optimization. The key insight, that $\omega$ effectively scales the learning rate, explains why constant $\omega$ (standard practice) may be suboptimal and motivates our schedule design. The empirical validation comes from Figure 4 and Table 5, where the proposed schedule consistently outperforms constant $\omega$ across all tested configurations. We would however like to note that the relation between a layers $\omega$ parameter and its effective learning rate $\tau$ is independent of the optimization step and applies generally.
>
> **(7) Hyperparameter sensitivity**: While there are subtle differences across datasets, the overall trends remain consistent. We note that hyperparameter sweeps are relatively inexpensive in practice, as early-stage performance reliably predicts late-stage results (training for just 2-3k iterations is typically sufficient to assess performance). However, we had consistently good results using the parameters reported in the paper.

---

> ### Author Response · Authors · 2026-01-22
> **Response to Reviewer pR1Y - Part 2**
>
> **(8) Interpolation between signal-specific parameters from different datasets:** Thank you for the interesting idea. Since this is not straightforward in the current setup, we will keep this in mind for future research.
>
> **Reporting compression ratio:** You are right, that COIN and COIN++ apply similar methods for image compression. We would like to clarify that compression, while being one potential application, was beyond the scope of this paper.
>
> **Comments on notation:** We would like to thank you for making us aware of the notation error on page 9. We corrected it in the revised version of the paper. We however don't understand the comment regarding the two $\phi$, as they appear to be identical. We carefully double checked our notation again and can't identify inconsistencies. Can you maybe clarify this, so that we can properly address the comment?
>
> *We would again like to thank you for this extensive review and are happy to engage during the discussion period.*

---

> ### Comment · Reviewer_pR1Y · 2026-01-26
> **Response to Rebuttal**
>
> I would like to thank the authors for their insightful and thorough answers. It is clear to me that this is a very well-thought-out paper which a lot of work went in, which is strongly reflected by the knowledgeable answers. I am happy to raise my score to a 4 given this rebuttal.
>
> The 3D qualitative results do leave me with an additional question. Though I understand the method's main merit is a generalizable strategy for extracting contextually meaningful latent vectors, I wonder if the 3D results in their current state (with $32^3$ resolution) could already offer some clinically meaningful interpretation.
>
> Furthermore, though I understand the point of training methods in similar conditions, it is still my opinion that some performance upper bound, even if trained by the authors themselves, would be a meaningful additional result. This could serve as an 'ideal' that you would like the latent vectors to achieve as some sort of near-lossless compression. Having both a baseline in comparable conditions and a more optimized one would benefit and not weaken the paper, in my opinion.
>
> P.S. It looks like the $\phi$ notation inconsistency was on me, I must have gotten confused with my remark on $\omega$. My apologies.

---

### Official Review · Reviewer_QtRf · 2026-01-06

**Confidence:** 4
**Preliminary Rating:** 5
**Final Rating:** 5

**Summary:**

This paper introduces MedFuncta, a meta-learning framework for unified neural fields (NFs) across diverse medical datasets. They follow the Funta paper’s idea [1] to represent each n-dimensional signal via a shared SIREN backbone and a single 1D latent vector and use the latent vector for downstream tasks. They propose two major engineering contributions: layer-dependent ω-schedule to increase the learning capacity of the SIREN network and context reduction via random subsampling to reduce memory consumption. Extensive experiments show strong reconstruction quality across multiple medical datasets and competitive performance on simple downstream classification. The authors also release code and weights and a large MedNF dataset.

Ref:
[1] From data to functa: Your data point is a function and you can treat it like one. In International Conference on Machine Learning, pages 5694–5725. PMLR

**Strengths:**

1. The two engineering innovations are solid and effective. The layer-dependent ω-schedule for the widely used SIREN backbone has shown effective results by encouraging staged optimization dynamics. And the context-reduced meta-learning framework via random subsampling is necessary to handle large data with reasonable GPU memory load.
2. The experiments are well designed and extensive, covering broad modalities (1D ECG, many 2D modalities, and 3D MRI/CT).
3. The well-maintained codebase, model weights, and a large MedNF latent resource are valuable to the community. I especially applaud the authors for sharing their negative results (Appendix H), which may greatly benefit further research.

**Weaknesses:**

1. The size of resolution of the data is still limited as the authors already acknowledged in the paper. It is unclear whether scaling this approach to high/realistic resolution will match the benchmark performance with reasonable computational overhead.
2. The practical value of a shared 1D representation is still vague. The paper motivates “consistent downstream processing across diverse data types”, but the only demonstrated downstream task is relatively simple classification. While conceptually appealing, it remains somewhat vague how this abstraction would extend to other tasks (segmentation, registration, inverse problems and generation) and would be used in real medical workflows to benefit clinicians. In particular, the learned 1D latent seems difficult to interpret, and the framework currently offers limited insight into what clinically meaningful factors they capture.
3. The paper argues benefits for “irregularly sampled, heterogeneous data”, but the showcased experiments are mainly on regular grids. More examples on these cases would better justify this part of the claim.

**Detailed Comments:**

NA

**Justification Of Final Rating:**

The authors have addressed my questions and other reviewers' questions well. And it is encouraging to see they are already planning the next steps to make this tool clinically useful. I will keep my original rating.

**Justification Of The Preliminary Rating:**

This work addresses an important challenge to extend meta-learning of neural fields to medical datasets of broader categories and higher resolution.
The study is comprehensive across modalities (1D/2D/3D), includes meaningful ablations, and the proposed ω-schedule and context-reduction are practical improvements with clear benefits.
I also appreciate the commitment to reproducibility. Releasing code, weights, and the larger MedNF dataset will likely make this work useful beyond the specific experiments.
My main remaining concern is the gap between the broad motivation (a unified, general-purpose 1D representation for heterogeneous/irregular medical data and many downstream tasks) and what is directly demonstrated, which is mostly regular-grid reconstruction plus relatively simple classification.

Overall, it is a well-written paper with comprehensive and convincing results and is worth presenting at the MIDL conference.

**Questions To Address In The Rebuttal:**

It would be great if the author could include some discussion on the following points:
(1)	What is the anticipated computation needed to scale up to some of the standard high-resolution data (e.g., 2D chest X-ray of 1024x1024 and 3D CT of 512x512x512)?
(2)	What are the realistic clinical workflows where a unified 1D latent would be beneficial for multimodal medical data compared to standard encoder embedding? A few concrete examples would help.

---

> ### Author Response · Authors · 2026-01-22
> **Response to Reviewer QtRf**
>
> We would first like to thank the reviewer for the very positive feedback and will address open questions about our paper in the following response.
>
> **Anticipated compute for further scaling the method:** Thank you for this relevant question that we are actively exploring. While it is difficult to provide specific numbers at this stage, we would like to share some ideas for reaching higher resolutions. The main limiting factor is indeed the GPU memory for storing the compute graph of the inner-loop optimization during training, while using large batch sizes for training stability. We believe the most promising direction is a combination of a reduced number of inner-loop steps together with a mini-batch training scheme, in which we sequentially run several inner-loop optimizations on smaller batches, aggregate the gradients, and then perform a single meta-update. This is mathematically equivalent, but results in a largely reduced GPU memory requirement and can efficiently be implemented in a distributed environment. We also think that novel optimization strategies we've worked on [1] will help to further scale our framework by explicitly optimizing networks for expressiveness.
>
> [1] McGinnis, J, et al. “*Optimizing Rank for Optimizing Rank for High-Fidelity Implicit Neural Representations.*” arXiv preprint 2025.
>
> **Clinical workflows that we could apply our method to:** We specifically intend this study to introduce our framework to the medical imaging community, leaving specific applications open to be explored (supported by our substantial open science contribution: code + first large-scale dataset). Our method can be applied to all kinds of problems in which NFs are supposed to generalize across a population/dataset including but not limited to segmentation, registration, image-to-image translation, atlas building, spatiotemporal modeling, and many more (we more explicitly highlight applications in the revised version of the manuscript). A specific application it has already been applied to is B-line and lesion classification, as well as ejection fraction prediction on variable length ultrasound videos. We hope to see more applications and are ourselves working on style-transfer and generative modeling within our proposed framework.
>
> *We would again like to thank you for your encouraging comments and are happy to answer further questions during the discussion period.*

---

### Official Review · Reviewer_5Jf5 · 2026-01-07

**Confidence:** 4
**Preliminary Rating:** 4
**Final Rating:** 5

**Summary:**

This paper introduce a novel method for training Neural Fields (NFs) on large medical datasets by representing medical signals as continuous functions rather than discrete grids. Each instance is encoded into a compact latent vector that modulates a shared, meta-learned NF, enabling efficient generalization across datasets. Experiments across diverse medical datasets show competitive performance with reduced memory and compute via sparse meta-learning. The released MedNF dataset supports future research in large-scale medical neural representations.

**Strengths:**

1. Propose a novel method for unifying different medical data modalities together, allowing generalization with a Signal-specific parameters, drastically improving computational efficiency and scalability

2. Introduced w-Schedule in training. Gradualy increased ω with network depth, achiveing better performance and fatser convergence than  state-of-the-art networks with a constant ω-parameter. Validated this theoretical insights through experiments.

3. Experimented with multiple dataset across many modalities, showing the effectiveness of the proposed method.

4. Conduected ablaton study to furthur validate the proposed context reduction strategy

**Weaknesses:**

1. The practical application of the proposed framework is insufficiently clarified, as the paper demonstrates strong reconstruction and representation learning results but does not clearly articulate which concrete medical tasks or clinical workflows MedFuncta is intended to directly support.
2. The model appears to be largely dataset-specific, as the data used to train the meta-learned shared parameters must come from the same dataset as the data used for test-time adaptation, which may limit the method’s ability to generalize to unseen datasets or real-world clinical scenarios without re-training.

**Detailed Comments:**

1. Additional clarification on the experimental training setup would behelpful. Particularly regarding which datasets are used to train the meta-learned shared parameters and how these relate to the data used during test-time adaptation.

2. Further explanation of the intended clinical use cases would strengthen the paper, as it is currently unclear which specific medical tasks or workflows would most directly benefit from the proposed method.

**Justification Of Final Rating:**

The paper proposes a novel method for generalizing neural fields to large medical datasets across multiple modalities. It is technically solid and supported by a substantial set of experiments and ablation studies. The author added clinical applications in the revised version and clarified my question regarding cross-dataset generalization.

**Justification Of The Preliminary Rating:**

The paper proposes a novel method for generalizing neural fields to large medical datasets across multiple modalities. It is technically solid and supported by a substantial set of experiments and ablation studies. However, the paper would benefit from a clearer articulation of its intended clinical applications and a more explicit discussion of dataset specificity and cross-dataset generalization.

**Questions To Address In The Rebuttal:**

1. How the meta-learning training is set up in the experiments, specifically which datasets are used to train the shared meta-learned parameters and whether the same datasets are used during test-time adaptation?
2. How does the proposed framework generalize to signals from unseen datasets or distributions, and would applying MedFuncta to a new dataset or modality require re-training the meta-learned shared parameters?

---

> ### Author Response · Authors · 2026-01-22
> **Response to Reviewer 5Jf5**
>
> We would first like to thank the reviewer for the positive feedback and will address all open questions in the following response.
>
> **Practical application:** We agree that highlighting potential applications and future research directions more explicitly would strengthen the paper. Our method is a learning paradigm applicable to many generalizable NF tasks, including those mentioned in our revised discussion/future work section. Specifically, it can be applied to tasks where NFs are expected to generalize across a population/dataset, and cases in which jointly modeling a distribution of NFs across a population is beneficial. It can also be applied in scenarios where training NFs on large-scale datasets is required and single-instance training becomes prohibitively expensive. Clinically relevant applications include segmentation, registration, image-to-image translation, atlas building, or spatiotemporal modeling (references in the revised version). Indeed, our method has already been applied to another domain, ultrasound videos, where ejection fraction is predicted from signal-specific parameter vectors fitted to videos of variable length. To accommodate your comment, we revised the discussion/future work section to describe potential applications and future research opportunities more explicitly.
>
> **Training setup & generalizability to unseen datasets:** This is a critical point of our work and we are happy to clarify this further. You are correct, we train one shared model per dataset. Following common practice, we train on a training set and evaluate on an unseen test set (details in Section 3). Our training setup is motivated by one fundamental design choice: the shared model learns transferable features about signal structure, smoothness, and local correlations for a given modality/anatomy, while the signal-specific parameters capture subject-specific features. Training models that generalize across multiple modalities and anatomical regions is something we plan to explore in the future, being particularly interested to see if it's possible to capture useful information about signals from different modalities.
>
> *If you have any further questions or comments, we would be more than happy to engage during the discussion period and hope our response addresses your open questions.*

---

> > ### Comment · Reviewer_5Jf5 · 2026-01-28
> >
> > Thank you for answering my question and addressing them in the revised version. I'll raise my score to 5.

---

### Author Rebuttal · Authors · 2026-01-22

**Rebuttal:**

We would like to thank all reviewers for their positive and motivating feedback. We are encouraged by your comments characterizing our work as *"addressing an important challenge"* and *"well-written with comprehensive and convincing results"* (QtRf), *"technically solid and supported by a substantial set of experiments"* (5Jf5), and acknowledging our *"significant open science contribution"* (pR1Y).

In response to your feedback, we have revised the manuscript (changes highlighted), and have added more details to clarify specific use cases and future directions for research. Moreover, we have conducted additional experiments, including a comparison to first-order methods, and have added additional qualitative results to the revised manuscript.

We kindly refer to our individual responses for detailed answers to specific reviewer questions.

*If you have further comments or questions, we'd be happy to actively discuss them during the discussion period.*

**Supporting Material:**

/attachment/dcafca23947a27ba13e1957de136bc7f270a7b4c.pdf

---

### Comment · Area_Chair_fzfL · 2026-01-28
**Engage in discussion & final rating reminder**

I would like to encourage reviewers to engage with the authors during the discussion phase to clarify any missing or contradictory points.

Please ensure that the final rating is updated by February 1st 2026 (23:59 AoE).

---

### Meta-Review · Area_Chair_fzfL · 2026-02-05

**Recommendation:** Accept (Oral)
**Confidence:** 4

**Metareview:**

All reviewers positively evaluated the contributions of this paper (two *Strong Accept* and one *Weak Accept*).

The authors propose a novel method and show promising experimental results, also discussing their limitations to low resolutions.

They also make a significant open-science contribution by releasing a MedNF dataset, code, trained models, and negative results, supporting transparency in the medical imaging community.

The authors have addressed most of the reviewer concerns during rebuttal, adding visualizations and further justification in the Appendix and revised manuscript.

---

### Decision · Program_Chairs · 2026-02-13

Accept (Poster)